# A trans-oceanic flight of over 4,200 km by painted lady butterflies

Tomasz Suchan [1,7], Clément P. Bataille [2,7], Megan S. Reich [3], Eric Toro-Delgado [4,5], Roger Vila [5], Naomi E. Pierce [6] & Gerard Talavera [4,6,7] ✉

The extent of aerial flows of insects circulating around the planet and their impact on ecosystems and biogeography remain enigmatic because of methodological challenges. Here we report a transatlantic crossing by *Vanessa cardui* butterflies spanning at least 4200 km, from West Africa to South America (French Guiana) and lasting between 5 and 8 days. Even more, we infer a likely natal origin for these individuals in Western Europe, and the journey Europe-Africa-South America could expand to 7000 km or more. This discovery was possible through an integrative approach, including coastal field surveys, wind trajectory modelling, genomics, pollen metabarcoding, ecological niche modelling, and multi-isotope geolocation of natal origins. The overall journey, which was energetically feasible only if assisted by winds, is among the longest documented for individual insects, and potentially the first verified transatlantic crossing. Our findings suggest that we may be underestimating transoceanic dispersal in insects and highlight the importance of aerial highways connecting continents by trade winds.

Long-range insect migration has long fascinated scientists[1–4]. Insects on the move are often observed far from their native range in open waters from ships at sea and from ocean platforms, on remote, unsuitable islands, or arriving along coastal beaches from trajectories that appear to stretch far out to sea[5–10]. Despite decades of accumulated evidence, insect long-distance dispersal (LDD) nevertheless remains overlooked due to the dearth of reliable methods to track long-distance movements of such small and short-lived organisms. Miniaturized VHF radio transmitters have been successfully used to study the navigation ability of large insects overnight[11] and radar tracking of insects is feasible over short distances[10,12,13]. While promising, these technological advances limit dispersal records to short timescales (<day), sites with preexisting infrastructure (e.g., radars), and, due to body mass requirements, large insects. Consequently, they are not scalable to most migratory insects and sampling locations,

potentially leading to an underestimation of the dispersal capacity of insects.

Here, we investigate a dispersal event by a flock of painted lady butterflies, *Vanessa cardui*, found on the Atlantic coast of South America (French Guiana), outside their native range. Three of about ten observed individuals were captured alive on the beach at ~6:00 am on the 28th of October 2013, apparently arriving after a vigorous flight across the ocean, judging from their damaged wings and resting behavior on the sand. Painted ladies are strong migrators, known for their recurrent trans-Saharan flights and a multigenerational cycle spanning ca. 15,000 km between the Afrotropical and the Palearctic regions[14–18]. *V. cardui* is nearly cosmopolitan, but stable populations have not been recorded from South America. The individuals found on the coast of French Guiana should therefore have originated from populations in North America, Europe or Africa. Using an integrative

[1]W. Szafer Institute of Botany, Polish Academy of Sciences, Kraków, Poland. [2]Department of Earth and Environmental Sciences, University of Ottawa, Ottawa, ON K1N 6N5, Canada. [3]Department of Biology, University of Ottawa, Ottawa, ON K1N 6N5, Canada. [4]Institut Botànic de Barcelona (IBB), CSIC-CMCNB, Barcelona 08038 Catalonia, Spain. [5]Institut de Biologia Evolutiva (CSIC-Univ. Pompeu Fabra), Barcelona 08003 Catalonia, Spain. [6]Department of Organismic and Evolutionary Biology and Museum of Comparative Zoology, Harvard University, Cambridge, MA 02138, USA. [7]These authors contributed equally: Tomasz Suchan, Clément P. Bataille, Gerard Talavera. ✉e-mail: gerard.talavera@csic.es

approach that combines multiple sources of evidence, we show that the individuals found in South America migrated across the Atlantic Ocean from West Africa, with probable origins traced back to Europe. This journey encompassed a minimum flight distance of 4200 km over the ocean, and potentially exceeding 7000 km from the point of butterfly emergence.

## Results

### Trans-oceanic wind trajectories

To investigate the dispersal history of the captured butterflies, we used a suite of methods that enabled the reconstruction of their natal origin and potential dispersal pathway. First, given the critical role of winds to assist insect LDD[19], we reconstructed hourly wind backward trajectories from the capture sites, using the Hybrid Single-Particle Lagrangian Integrated Trajectory (HYSPLIT) dispersion model from NOAA's Air Resources Laboratory. We inferred wind trajectories at different altitudes (500 m, 1000 m and 2000 m.a.g.l) over a 200 h period (~8.3 days) for a range of days both before and after the capture date of October 28th (Supplementary Fig. S1). Trajectories were inconsistent in directions and altitudes for the five days before the 26th of October (21st–26th October) and the three days following the capture date (28th–31st), with only a minority of trajectories originating in Africa (Supplementary Fig. S1 and Supplementary Table S1), and all limited to one altitudinal layer. Conversely, the trajectories for the 48 h immediately preceding the capture date (26–28th October) were consistent at different altitudes and exceptionally favorable for the butterflies to disperse across the Atlantic from West Africa, assisted by winds (Fig. 1 and Supplementary Fig. S1). 83% of these trajectories crossed the West African coastline, at an average of 159.2 ± 1.28 (SE) hours (ca. 6.6 days) since origin (Supplementary Table S1 and Supplementary Fig S1). Mean windspeed for all computed trajectories between 26–28th of October was 7.47 ± 0.07 (SE) m/s (ca. 26.89 km/h) (Supplementary Table S2 and Supplementary Fig. S1).

### Genetic assignments to source populations

Second, we analyzed genome-wide genetic diversity from the South American samples and compared them to 126 *V. cardui* individuals from three continents, including 40 from North America, 56 from Europe and 34 from Africa (Supplementary Data 1 and Supplementary Fig. S3). Phylogeographic analyses using RAD-sequencing showed genetic differentiation between North American and European-African populations, thus allowing assignments of the intriguing South American samples. Principal component analysis, as well as co-ancestry analysis unambiguously showed a clustering of the South American samples within the European-African population (Fig. 2). This result ruled out the possibility that the migrants originated in North America.

### Plant distributions of transported pollen grains

Third, we isolated DNA from pollen grains found on the bodies of the butterflies and sequenced the plant internal transcribed region 2 (ITS2) of the pollen through a metabarcoding approach[20]. The resulting sequence reads were classified to the species level and recovered a mixture of pollen from 8 to 15 species of plants, depending on the classification method (Fig. 3 and Supplementary Table S3). Most of the identified plant species had a Neotropical distribution or were poorly informative due to their widespread distribution, but a well-represented plant (by number of reads) was the Sahelian endemic *Guiera senegalensis* (73–98% of the total sequence reads classified with probability of 95% or higher, depending on a bioinformatic pipeline). To a lesser extent, another sub-Saharan species, *Ziziphus spina-christi* was also detected (Fig. 3, Supplementary Table S3). Both species are shrubs flowering at the end of the rainy season in West Africa, from August to November, and are therefore meaningful candidates as nectar sources for the dispersing butterflies. Their Sahelian

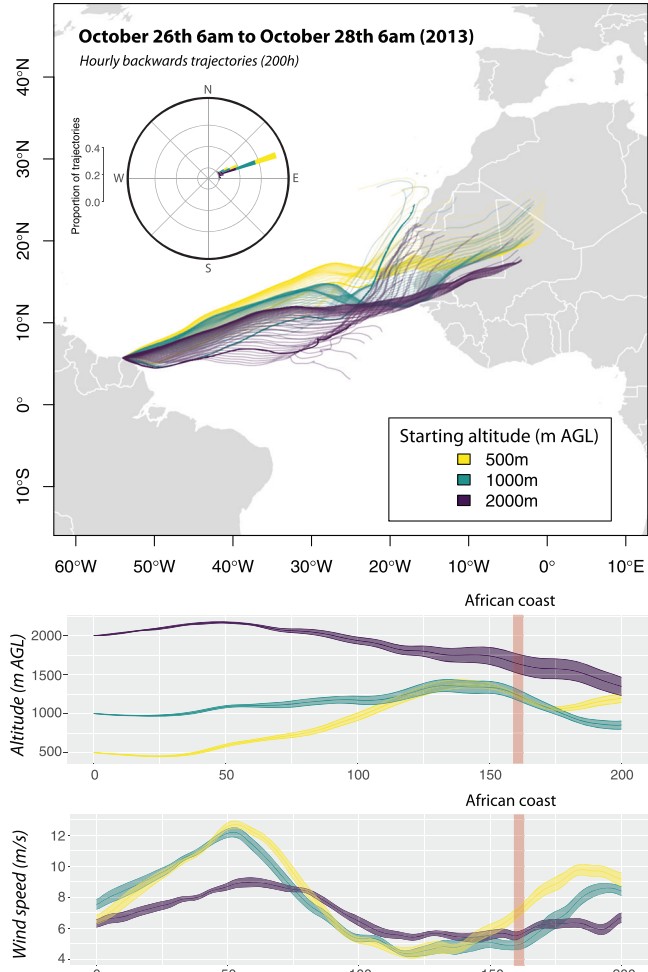

**Fig. 1 | Wind backtracking suggests the transoceanic route followed by the painted lady butterflies.** Hourly backward wind trajectories for the 48 h prior to the butterfly flock observation at 6 am on October 28th, 2013. Trajectories were inferred by the Hybrid Single-Particle Lagrangian Integrated Trajectory (HYSPLIT) dispersion model[60], based on the Reanalysis database and computed on 200 h back trajectories for three different altitudinal layers (500 m, 1000 m and 2000 m.a.g.l.). Average altitude and speed ±SE along the course of the trajectories are represented for each altitudinal layer in the lower plots. Source Data can be obtained online using the provided code (see code availability).

distribution at the western African coast narrows the potential origin of the transatlantic flight.

### Geolocation of natal origins

Finally, we geolocated the natal origin of the mysterious individuals sampled in South America using stable and metal isotopes. We used a dual isotope approach, analyzing hydrogen ($\delta^2$H) and strontium ($^{87}$Sr/$^{86}$Sr) isotopes in the wings of each individual[18,21]. We found similar isotopic signals in each individual, suggesting that these butterflies had the same natal origin (Supplementary Table S4). We then compared these isotope signals to existing baseline isotope maps (i.e., isoscapes)[22–26], identifying possible regions of natal origin within the Afro-Palearctic range (Fig. 4A). To better delimit areas with highest probabilities, we overlaid isotope assignment maps with suitability maps of reproductive habitat for September and October, as recently inferred through ecological niche modeling (ENM)[16]. The combined isotope assignments show high densities of high-probability pixels in suitable regions of Western Europe (Portugal, France, Ireland and UK) and West Africa (Mali and coastal regions of Senegal and Guinea-

Bissau) (Fig. 4B), when the collected individuals would have been developing (given an approximate development time of 3 weeks and an adult lifespan of 3–4 weeks). Isotopes also signaled regions in North Africa, particularly in Tunisia and Libya, but we consider this assignment less likely given that the breeding habitat for *V. cardui* in this drier region only becomes suitable for the development of *V. cardui* later in the season[15,16] (Fig. 4).

## Duration and speed of trans-oceanic dispersal

We assessed the feasibility of a transatlantic crossing by estimating energetic requirements and dispersal duration of *V. cardui* when using different flight strategies (Supplementary Table S6). In the absence of wind-assistance, we estimate that painted ladies could travel a maximum of ~780 km without refueling, far less than the 4200 km distance across the Atlantic. Therefore, the painted ladies must have relied on the easterly trade winds that were present preceding the capture date. Furthermore, even with wind-assistance, painted ladies using an exclusively active flight strategy would travel a maximum of ~1900 km before depleting their energy reserves. Therefore, painted ladies must be using an alternating strategy of active flight and minimum-effort flight (i.e. flapping only to stay aloft and gliding), a behavior that is known from monarchs and other butterflies[27]. Assuming that painted ladies use the same alternating flight strategy as monarchs (with a 15:85 proportion of active:minimum-effort flight) and with the assistance of wind (average windspeed of 7.47 m/s based on trajectories starting 26–28th of October), the painted ladies we captured in French Guiana could have crossed the Atlantic from West Africa in 5–8 days, but only if their starting fat reserves were at least as high as 13.70% of their body mass.

## Discussion

Our interdisciplinary, multi-tool approach supports the hypothesis that the butterflies collected on the coast of French Guiana were present as adults in sub-Saharan Africa when accidentally caught up in a windborne, transatlantic dispersal event (Fig. 4). These could have

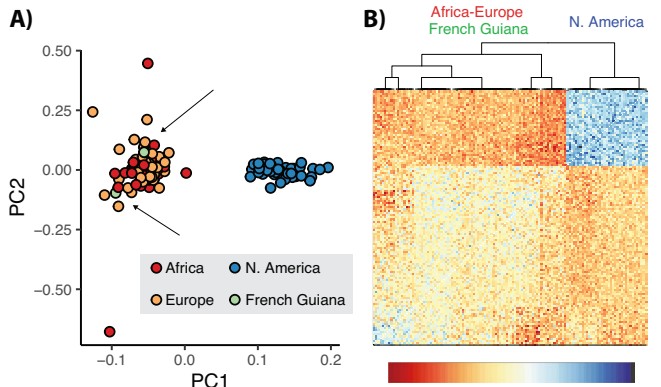

**Fig. 2 | Genome-wide phylogeographic assignments from SNPs obtained with a ddRAD sequencing approach demonstrate an Africa-Europe origin instead of a N. American one. A** Principal Component Analysis (PCA) using SNPs with less than 10% missing data per sample and pruned for LD (13 206 SNPs), the variances explained by the two first axes are 6.26% and 5.21%; **B** Co-ancestry matrix using fineRADstructure on a dataset of 7982 RAD loci. Both methods delimit two populations, one in North America and the other in Africa-Europe. Butterflies collected in French Guiana cluster with the Afro-European population. Source Data are provided as a Source Data file.

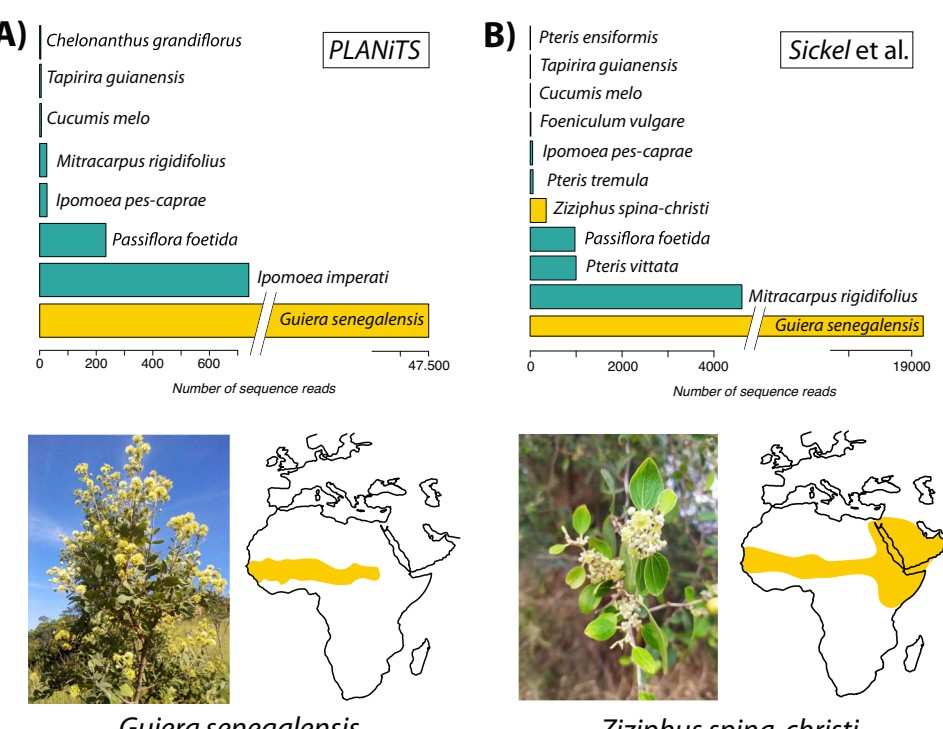

**Fig. 3 | Metabarcoding of the pollen carried by the butterflies indicates the African origin of the transatlantic trip.** Classification of the obtained ITS2 metabarcoding sequences processed using a denoising pipeline (see Methods), and blasted on curated databases from **A** PLANiTS[83] and **B** Sickel et al.[82] using the SINTAX classifier. In addition to plants present in French Guiana or widely distributed (green bars), two Sahelian endemic plants (yellow bars) were found among the pollen recovered from the bodies of the painted lady butterflies in South America: *Guiera senegalensis* and *Ziziphus spina-christi*, the former being especially common. Source Data is available in Supplementary Table S3.

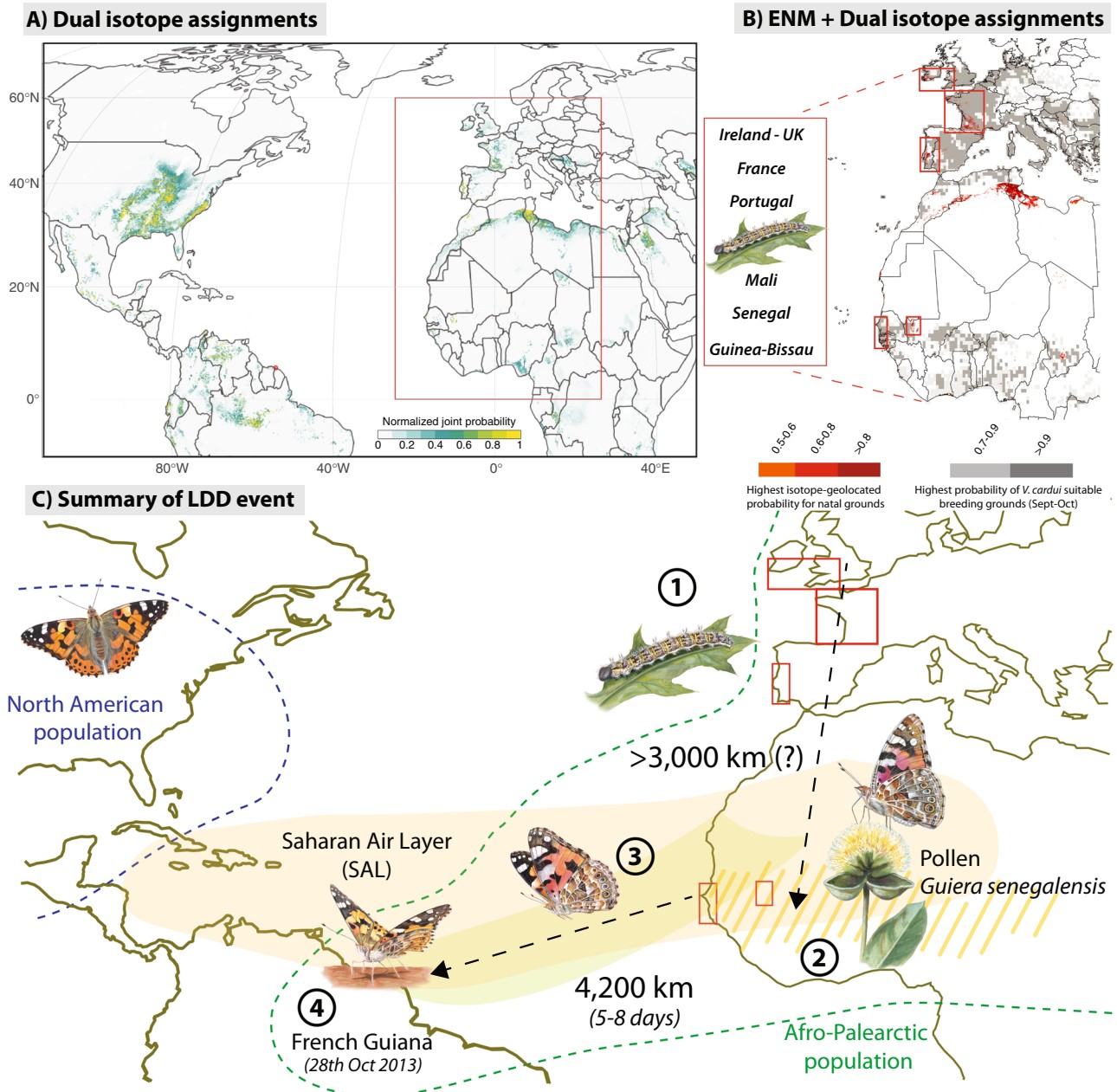

**Fig. 4** | *Vanessa cardui* **transoceanic dispersers likely developed as larvae in Western Europe. A** Natal grounds inferred by a dual approach using $\delta^2$H and $^{87}$Sr/$^{86}$Sr isotopes. Color scale depicts normalized joint probabilities by pixel. Assignments show high densities of high-probability pixels in America: Eastern North America, Venezuela, Brazil; Palearctic: Portugal, North Africa (mainly Tunisia and Libya), Saudi Arabia; Afrotropics: Mali, Nigeria, Chad and Congo. **B** Dual isotope assignments (in red, prob >0.5) overlaying with maps of highest probability breeding grounds for *V. cardui* inferred by Ecological Niche Modeling (ENM) for September and October (in gray, prob >0.7) in Southwest Palearctic and West Africa. Low suitability of reproductive habitat discards isotope-based predictions from North Africa, except a few pixels in coastal Morocco. **C** Infographic summarizing the possible natal grounds and dispersal pathway of a flock of *V. cardui* butterflies across the Atlantic from West Africa to South America, through a non-stop flight of a minimum of 4200 km during 5–8 days. The total flight distance for these individuals could be as long as 7000 km if they developed in Western Europe. Source Data can be obtained online using the provided code (see code availability). Butterfly illustrations by Blanca Martí.

developed as larvae in Western Europe and thus migrated to sub-Saharan Africa as part of their typical migratory cycle[14–17]. The combination of methods used in this analysis was able to uncover multiple details of the event and should be of similar utility in tracking other cases of insect LDD. While none of the five methods is definitive by itself, they complement each other and produce a coherent reconstruction of the course of the events. For example, isotope-based geolocation indicates zones with high probabilities of natal origin in the Eastern USA, which could be discarded using the results of genomic clustering and the presence of Afrotropical pollen. On the other hand, wind trajectories, pollen identification and genomics alone would not be enough to suggest a potential European natal origin without the isotope data.

Beyond the methodological advance that our integrative approach represents, this finding contributes to the understanding of the dispersal capacity of insects, both physiologically and biogeographically. How can such small insects cross an ocean without refueling? And how frequent are long-distance dispersal events? Extreme endurance in LDD by insects is not well understood. The models show that the transatlantic crossing was only possible with the

help of sustained easterly winds and with an important fraction of minimum-effort flight. We estimate that the journey was completed in 5–8 days. Given energetic constraints, we estimated that only the most fit butterflies, with high amounts of stored fat, and under favorable wind conditions, would have been able to complete the transatlantic journey, without the possibility to stop and feed. With lower energy reserves, they would have likely perished at the sea.

The correlation of wind currents with LDD dynamics in insects is known to be strong[10,19]. Nevertheless, it remains unclear to what extent recurrent winds at a global scale may function as dispersal highways and, thus, whether windborne insect dispersal may be predictable[28]. Every year, millions of tons of dust are transported from Africa to America across the Atlantic (i.e., the Saharan Air Layer - SAL) by the easterly trade winds, which has strong impacts on biogeochemical cycles and nutrient dynamics in the Caribbean and the rainforest soils of the Amazon basin[29–31]. The atmospheric transport of Saharan aeolian material occurs year-round, but has strong seasonal variations in dust concentration, altitude, and average trajectories[32]. In French Guiana, dust deposition generally peaks in spring, but occurs all year long[30]. This atmospheric pattern suggest the existence of a one-way dispersal highway for insects associated to the easterly trade winds.

Occasional dispersal of insects using the SAL is more likely to take place from July to November, when insect populations explode after the rains in the semiarid areas of Western Africa[17]. This is also the time (September-November) when southward migrations and extensive breeding of *V. cardui* occur in the region[14–18]. Despite the potential of the SAL as an aerial highway for *V. cardui* to reach the Americas, stable populations of this butterfly have never been found in South America and the Eastern Caribbean[33], suggesting niche limitations or species interaction incompatibilities. Rare migrants from Africa are most likely to arrive in the Guianas and the Eastern Caribbean, where only occasional occurrences of *V. cardui* have been documented[33–35]. For example, among the 26 observations of *V. cardui* along the South American coast and the Eastern Caribbean islands recorded to date on iNaturalist, 25 were documented between October and December. These instances align with the timeframe when *V. cardui* is highly present in West Africa, and suggest that dispersal from Africa may be more frequent than a singular exceptional event.

The LDD event by painted lady butterflies documented here covered at least 4200 km across the ocean and is among the longest demonstrated to date for individual insects. With the possibility of an origin in Western Europe, this journey could extend to 7000 km or more – a remarkable feat for a small-sized organism weighing less than a gram. A few other insects are also known to be able to endure LDD. For example, the dragonfly *Pantala flavescens* apparently migrates annually across the Indian Ocean[36]. The monarch butterfly (*Danaus plexippus*) also annually migrates between Canada and Mexico, and tagged individuals have demonstrated flights as long as 4635 km (2880 miles)[37,38]. Recent work using light aircraft and individual radio tracking of death head's hawkmoths (*Acherontia atropos*) recorded a remarkable maximum ground speed of 69.7 km/hour (19.4 m/s)[11]. These scattered reports of individual feats of migration both in terms of distance covered and flight speed are important: collectively they indicate that trans-oceanic LDD events may be sufficiently frequent to have played an underestimated role in biogeographic dispersal over time (cf. panbiogeography[39,40]).

The biogeographic distributions and phylogenetic histories of many insects could hardly be explained without long-range trans-oceanic dispersal[41–46]. The occurrence of *V. cardui* in Hawaii and a few historical records of the species from Western Australia[47,48] suggest other extreme trans-oceanic journeys, likewise the disjunct Holarctic distribution of its closely related species *Vanessa atalanta*[49] or the established American populations of the Old World tropical butterfly *Hypolimnas misippus*[50]. Larger insects, such as African desert locusts (*Schistocerca gregaria*), have also been reported in the Caribbean[51–53],

and westward dispersal from Africa have triggered the diversification of this genus in the New World[41,53]. Overall, and despite high intrinsic risks, LDD in insects may also entail a relevant rate of success even for the most extreme trans-oceanic crossings.

With global change, environmental triggers might increase LDD in insects, and excessive LDD of allochtonous, potentially invasive elements to ecosystems has been identified as one of the main threats to global biodiversity[54]. Droughts in Africa increase concentrations of subsequent transported aerial dust[29], and windscape connectivity has also been predicted to change in the future and to severely impact wind-dispersed species[55]. Importantly, insects are also known to respond to environmental stress with increased movement behavior[47,56], and migratory movements may increase the probability of accidental LDD. Ongoing shifts in range and dispersal patterns could not only reconfigure the distribution of individual species, but also those of their interactions with host plants and pathogens[56–58]. A better understanding of how extrinsic (e.g., windscape, climate, pollution) or intrinsic (e.g. physiology, plasticity, metabolism) factors limit or favor insect dispersal will help predict their potential ecological consequences. Implementing systematic assessments of LDD as standard biomonitoring routines should help to mitigate biodiversity threats stemming from global change, and we show that coastal areas are excellent sites to monitor overseas dispersal of insects.

## Methods

### Field sampling

A few occasional *V. cardui* records have been previously reported in the northern part of the South American Atlantic coast and the Caribbean in October[33–35]. Given that autumn coincides with southward migrations of *V. cardui* in the northern hemisphere (both in North/Central America and in Europe/Africa), we hypothesized that sampling in the autumn would increase our chances to collect long-distance immigrants. To this goal, coastal sampling surveys in French Guiana were conducted for several days in October 2013. Surveys usually took place at sunrise, starting around 6 a.m. until about 10 a.m. On October 28th, 2013, *V. cardui* butterflies were found with open wings standing in the sand, just a few meters from the water line. Three of about ten observed butterflies were collected and stored in sealed glassine envelopes for further molecular, isotope and pollen analyses (samples 14M973, 14M974, 14M975; Supplementary Data 1). For phylogeographic assignments, we gathered molecular data from samples in North America, Europe and Africa curated at the collections of the Institute of Evolutionary Biology (IBE, CSIC-UPF, Spain), the Botanical Institute of Barcelona (IBB, CSIC, Spain), and the Museum of Comparative Zoology (Harvard University, USA) (Supplementary Data 1). Samples used in this study were collected legally following local requirements. Specimens from French Guiana were collected before enforcement of the Nagoya protocol and outside of protected areas. Our study did not require ethical approval.

### Wind trajectories

Wind trajectory analyses have been instrumental in modeling long-distance movements of migratory insects[59,60]. We reconstructed backwards wind trajectories using the Hybrid Single-Particle Lagrangian Integrated Trajectory (HYSPLIT) dispersion model from NOAA's Air Resources Laboratory[61]. Analyses were based on the Reanalysis database and computed on 200 h from the coordinates of the observation site, for every hour between the 21st to the 31st of October. Trajectories were inferred for three different altitudinal layers (500 m, 1000 m and 2000 m.a.g.l.), thus covering a wide altitudinal range where migrant Lepidoptera are documented to move downwind[62–64].

We used a locally installed version of HYSPLIT and a custom R pipeline to automatically call multiple runs of the dispersion model. The script reads user-defined input parameters (coordinates, timeframe in hours, starting and ending hour, years, months and days on

which to run trajectories, duration and altitudes of trajectories and time zone) to generate a list of dates for which trajectories will be run using the *lubridate* R package[65]. First, the necessary meteorological files are downloaded from the Reanalysis database using the *splitr* R package[66]. Trajectory analyses, data transformations and plots are then performed for the required timeframes and altitudes. For each combination, the trajectories are computed using the R package *opentraj*[67], which builds and sends the HYSPLIT commands and parses its output into an R dataframe. This dataframe is then transformed into a "SpatialLinesDataFrame" object from the R package *sp*[68,69], which is used to plot the trajectories (Fig. 1, Supplementary Fig. S2). To generate the circular histograms, trajectory azimuths were calculated with the R packages *raster*[70] and *geosphere*[71] and plotted with *ggplot2*[72]. Both trajectory plots and circular histograms were generated at time ranges of 24 h and 48 h starting at 6 a.m. (UTC – 3 Time Zone), coinciding with the observation time in French Guiana on the 28th of October of 2013. Trajectories for the 48 h prior to the observation date and time are shown in Fig. 1. The altitudinal profile plot was generated with the results for the trajectory starting at the moment of the *V. cardui* observation (i.e., October 28th, 2013 at 6 a.m.). Palettes for all plots were obtained from the R package *viridis*[73]. Then, to infer times and percentages of backward trajectories reaching the African coast, we obtained shapefiles of the trajectories and calculated their intersection with a vector of the African coastline using the *terra* R package[74]. Lastly, we obtained windspeeds from HYSPLIT trajectories. The mean windspeed was determined by dividing the sum of distances by the total duration of each trajectory (200 h) (Supplementary Table S2). Because wind trajectories are expected to be variable over time, we further computed average windspeeds for each hourly segment within the trajectories, and calculated the mean of these segments across all trajectories (Fig. 1 and Supplementary Fig. S1). Analyses were conducted using R version 4.1.2, and R code is provided at the GitHub repository https://github.com/GTlabIBB/Guyane[75].

## Phylogeographic assignments

To test a phylogeographic relationship of the French Guiana specimens with their potential origins in other regions, we obtained ddRAD genomic libraries for two samples (14M973 and 14M975) plus 128 individuals in three continents: North America (40 individuals), Africa (34 individuals) and Europe (56 individuals). We prepared ddRAD libraries according to Peterson et al.[76], using EcoRI and BfaI restriction enzymes and size selection of 300 bp (280–320 bp) using a PippinPrep instrument and 2% agarose casettes (Sage Science, Beverly, MA, USA). These libraries were then sequenced in 150 bp, paired-end reads on Illumina HiSeq2500 at the Harvard University Bauer Core Facility. Raw reads were processed using ipyrad v.0.9.81[77] by mapping on *V. cardui* chromosome-level genome assembly (v.2.1)[78]. We used standard settings, except the minimum required number of samples per locus (min_samples_locus parameter) that was set to 65 (>50% of the samples). Detailed sample statistics are available in Supplementary Data 2 (number of raw reads, number of reads passed filters, number of mapped and unmapped reads, number of clusters and high-coverage clusters per sample, estimated heterozygosity and error rates, consensus clusters in each sample and the final number of loci in the assembly). Four samples were removed from the final dataset due to high proportion of missing data: 15R020, 15J795, 15O446, 15O450 thus the final dataset consisted of 126 individuals with 8337 loci containing 461,112 SNPs and 44.03% missing sites.

We resolved *V. cardui* population structure using fineRADstructure v.0.3.1[79] and PCA as implemented in PLINK v1.90[80]. For fineRADstructure, we exported the VCF file using finerad_input.py included in fineRADstructure-tools (https://github.com/edgardomortiz/fineRADstructure-tools), with a filter of a minimum of 2 samples in a locus, for 7984 RAD loci in total. As the PCA is sensitive to missing data[81], we filtered-out the sites with more than 10%

of missing data using VCFtools v0.1.16[82] resulting in 27,745 sites and then pruned linked sites in a window of 50 Kb, step size of 10 bp, and $r^2$ threshold of 0.1, which resulted in 13,206 sites retained for the PCA analysis. To control for the effects of missing data, we have plotted the results of the PCA *vs* the proportion of loci present in each sample (Supplementary Fig. S4). The North America – Europe/Africa divide was not driven by the proportion in missing data, despite some outlier samples with fewer shared loci in the African-European group.

## Metabarcoding of carried pollen grains

The analysis of pollen grains by the butterflies was performed by metabarcoding the internal transcribed region 2 (ITS2) according to the metabarcoding protocol of Suchan et al.[20]. Pollen was collected by vortexing the butterfly body (with wings removed) in a 2 ml tube with 50 μl in PCR-grade water with 0.1% SDS, centrifugation, removing the insect body, and drying the obtained solution under vacuum. The pellet was resuspended in 15 μl of Phire Plant Direct Buffer (Thermo Fisher Scientific, Waltham, MA, USA) and homogenized using zirconium beads for 1 min at 30 Hz in a TissueLyser II instrument (Qiagen, Hilden, Germany). First PCR, using ITS-S2F[83] and ITS-4R[84] primers tailed with Illumina adapter sequences, was conducted using 1 μl of the disrupted pollen sample, 25 μl of Phire Plant Direct Polymerase Mix, and 0.5 μM of each primer in a 50 μl reaction volume with the following PCR program: 98 °C for 5 min, 20 cycles of denaturation at 98 °C for 40 s, annealing at 49 °C for 40 s, elongation at 72 °C for 40 s, and a final extension step at 72 °C for 5 min. Each sample was processed in three replicates which were combined after the reaction, purified using 1:1 proportion of AMPure XP (Beckman Coulter, Brea, CA, USA) to the reaction volume, and eluted in 10 μl of water. The purified product of the first PCR was then used in an indexing PCR reaction consisting of 1 μl of the purified PCR product, 0.4 U of Q5 Hot Start Polymerase (New England Biolabs, Ipswich, MA, USA), 1× Q5 buffer, and 0.5 μM of each primer in a 10 μl reaction volume with the following PCR program: 30 s initial denaturation at 98 °C, 12 cycles of denaturation at 98 °C for 10 s, combined annealing and extension at 72 °C for 30 s, and final extension at 72 °C for 5 min. Again, each sample was processed in triplicates, which were pooled and purified with AMPure XP as above. All primer sequences are provided in Suchan et al.[20].

The resulting DNA libraries were sequenced on Illumina MiSeq (San Diego, CA, USA) using 600-cycle MiSeq Reagent Kit v3. We merged the resulting paired-end reads using PEAR v0.9.6[85], removed the primer sequences with CUTADAPT v3.2[86], and filtered reads with VSEARCH v2.20.0[87] by: expected error rate of 1, minimum length of 250 nt, without ambiguous nucleotides, and present in more than one copy. In order to account for the differences resulting from different bioinformatic treatment of the data, the merged reads were subject to three different treatments: i) clustering at 98, and ii) 99% sequence similarity, and iii) denoising using UNOISE algorithm[88] implemented in VSEARCH resulting in zero-radius OTUs (ZOTUs), all followed with de novo chimera removal. The processed reads were classified using SINTAX tool[89] in VSEARCH using two different reference databases: i) Sickel et al.[90] and ii) the PLANiTS database[91]. We then discarded sequences matching to the algae present in Sickel et al.[90] database and removed sequences classified as *Alternanthera* sp., as this is an erroneous GenBank entry (JX136744.1), most likely of fungal origin. The number of reads classified to each species with probabilities greater than 95% are presented in Supplementary Table S3. A comprehensive R script is provided at the GitHub repository: https://github.com/GTlabIBB/Guyane[75].

## Isotope-based geographic assignment

Stable isotopes have become effective tools to track the long-distance movements of insects. As insects mature, they incorporate local isotopic signatures from their food and drink into their developing

tissues. Tissues that have low metabolic activity, like insect wings, preserve these natal isotopic signatures[92,93]. Thus, the isotopic signature of a dispersing insect's wing can be measured and compared to a spatial model of isotopic variation (i.e., an isoscape) to estimate the insect's area of natal origin, a process termed geographic assignment[94]. In this study, we use hydrogen isotope values ($\delta^2$H) and strontium isotope ratios ($^{87}$Sr/$^{86}$Sr), which have been shown to be excellent complements to each other because they have different drivers of spatial isotopic variation (i.e., variation in $\delta^2$H is mainly driven by precipitation and variation in $^{87}$Sr/$^{86}$Sr is mainly driven by bedrock geology)[21]. The combination of these two isotopes in a dual approach typically results in a more spatially precise estimate of natal origin[18,21].

All modeling was performed in R language version Rx64 4.3.2 (https://www.r-project.org/) and the R script used to generate the isoscapes and probability surfaces is provided at https://github.com/GTlabIBB/Guyane[75].

**Isoscapes.** Hydrogen isotopes fractionate during trophic interactions and physiological processes, resulting in distinct isotopic composition among species and tissues[95]. We generated a $\delta^2$H isoscape for butterfly wings following the procedure described in the *assignR* package[96]. The process requires a tissue-specific isotope dataset of known-origin individuals to develop a calibration equation between a precipitation isoscape[24] and the tissue of interest. We compiled a $\delta^2$H known-origin dataset of 130 monarch butterflies from 31 sites across North America, already assembled in the *suborigData* object in the *assignR* package[23,97,98], and 142 butterflies from 56 sites across Europe and Africa[26]. When plotted against the global mean-annual precipitation $\delta^2$H isoscape from the *assignR* package, we obtained the following calibration equation: wing $\delta^2$H = 0.66 × precipitation $\delta^2$H – 49.34 ‰ ($R^2$ = 0.55). We calibrated the butterfly wing $\delta^2$H isoscape using the *calRaster* function in the *assignR* package (Supplementary Fig. S4).

Unlike $\delta^2$H, $^{87}$Sr/$^{86}$Sr do not fractionate across trophic levels, and the $^{87}$Sr/$^{86}$Sr of butterfly wings are similar to the bioavailable $^{87}$Sr/$^{86}$Sr of their natal environment[99]. Thus, we used the global bioavailable $^{87}$Sr/$^{86}$Sr isoscape from Bataille et al.[22] (Supplementary Fig. S4) to estimate the natal origin of the *V. cardui* samples based on their $^{87}$Sr/$^{86}$Sr ratios. This isoscape was modeled by coupling known-origin bioavailable $^{87}$Sr/$^{86}$Sr data with spatial biogeoenvironmental covariates within a random forest regression framework to predict the spatial variation in $^{87}$Sr/$^{86}$Sr from local plants, soils, and animals[22].

**Hydrogen isotope composition analysis.** Sample preparation and hydrogen isotope analysis were conducted at IsoForensics, Inc. in Salt Lake City, Utah, USA. Before measuring the $\delta^2$H values of the wings of each of the three *V. cardui* butterflies, we cleaned the wings in a 2:1 chloroform:methanol solution to remove lipids, dust, and contaminates from the wings that could impact the $\delta^2$H signature of the wing samples. For each individual, we sampled a piece of the cleaned wings in the same lower distal portion of the wing. The $\delta^2$H value of the non-exchangeable hydrogen of butterfly wing was determined using the comparative analysis approach described by[100] based on two calibrated, powdered keratin (horn) hydrogen-isotope reference materials (DS: −174.1 ± 3.4‰, ORX: −35.4 ± 4.1‰)[97]. Samples and standards were equilibrated with laboratory air for a minimum of 3 days prior to weighing. Samples and standards were then weighed into silver capsules (100 ± 10 µg) and dried under vacuum for a minimum of 5 days prior to isotope analysis. We performed hydrogen isotopic measurements on $H_2$ gas derived from high-temperature (1400 °C) flash pyrolysis of wing samples and keratin standards. Resultant separated $H_2$ was analyzed on an interfaced Thermo Finnigan MAT 253 isotope ratio mass spectrometer with a Thermo high-temperature conversion elemental analyzer (TC/EA; Bremen, Germany) attached via a ConFlo IV interface (Thermo Finnigan, Bremen, Germany). Our analysis assumes

that the fraction of exchangeable H atoms is similar between the keratin standards and chitin wing samples (i.e., -10%), and that the non-matrix matched standards (keratin vs chitin) and differences in preparation (powdered standards vs cut samples) do not bias the analysis. All $\delta^2$H values are reported to the international scale VSMOW-SLAP. Measurement of the two keratin laboratory reference materials corrected for linear instrumental drift were both accurate and precise, with typical within-run measurement error (1 SD < 2‰). An additional keratin standard, POW, was used as a quality check. The measured values for POW (−102.2 ± 1.26‰, $n$ = 4) were within the reported value and uncertainty ($\delta^2$H = −101.1 ± 3.0‰), thus verifying this approach. Analytical precision of these measurements is based on the reproducibility of POW, and is better than ±2‰.

**Strontium isotope ratio analysis.** A single wing from each of the three *V. cardui* butterflies was used to measure $^{87}$Sr/$^{86}$Sr in the wing. To remove surficial contaminants, wing samples were dry-cleaned with pressurized nitrogen gas for 10 min at about 10 psi. Samples were then digested in concentrated nitric acid (16 M; distilled TraceMetal™ Grade; Fisher Chemical, Canada) for 15 min at 250 °C using microwave digestion (Anton Paar Multiwave 7000, Graz, Austria). After digestion, the vials were limpid suggesting complete digestion. After drying, the sample was treated with 1 ml of 16 M $HNO_3$ and then transferred to a 7 ml Savillex PTFE vial. A 50 µl aliquot of the resulting solution was then analyzed for Sr concentration using ICP-MS (Agilent 8800 triple quadrupole mass spectrometer) at the Department of Earth and Environmental Sciences, University of Ottawa. Single element certified standards (SCP Science, Montreal, Canada) were prepared as calibration standards. Subsequently, the remaining sample was dried down and re-dissolved in 1 ml of 6 M $HNO_3$. A 100 µl microcolumn loaded with 100–150 µm Sr-spec Resin™ (Eichrom Technologies, LLC) was used for the separation of Sr. The matrix was rinsed with 6 M $HNO_3$, and Sr was collected with 0.05 M $HNO_3$, dried, and re-dissolved in 200 µl 2% v/v $HNO_3$ for $^{87}$Sr/$^{86}$Sr analysis.

The $^{87}$Sr/$^{86}$Sr analysis was conducted using a Nu-Plasma II high-resolution MC-ICP-MS (Nu Instruments) coupled to a desolvating nebulizer (Aridus II™, CETAC Technologies) at the Pacific Centre for Isotopic and Geochemical Research, University of British Columbia. Strontium isotope compositions are reported as $^{87}$Sr/$^{86}$Sr and normalized for instrumental mass bias using $^{86}$Sr/$^{88}$Sr of 0.1194. Isobaric interferences were corrected by measuring $^{83}$Kr and $^{85}$Rb. The means of 5 ppb NIST SRM987 and 1.4 ppb NIST SRM987 were 0.71025 ± 0.00009 (1 SD, $n$ = 138) and 0.71019 ± 0.00011 ($n$ = 48), respectively. To match the samples, a chitin internal standard, 5 ppb Alfa Aesar chitin (0.713959 ± 0.00009, $n$ = 3), was also used.

**Posterior probability surfaces.** We used the continuous-surface probability framework and the *pdRaster* function from the *assignR* package[96] to estimate the most likely locations of origin of each individual using single ($\delta^2$H or $^{87}$Sr/$^{86}$Sr) and dual ($\delta^2$H and $^{87}$Sr/$^{86}$Sr) isotope data. As the isotope signals from the different individuals are nearly identical, we assumed they were from the same region of natal origin and collated the probability map of each individual using the *JointP* function in the *assignR* package[96].

**Energetic flight models**

Given the exceptionally long dispersal event across the Atlantic by *V. cardui* butterflies, we investigated the energy requirements of different flight strategies, including (1) active flight in the absence of wind assistance, (2) active flight in the presence of wind assistance, and (3) an alternating strategy of active and "minimal-energy" flight phases. During active flight, painted ladies can achieve airspeeds of 6 m/s in the absence of wind[101]. We estimated an average mass of 150 mg for *V. cardui*, based on a dataset of lab-reared, ten-day-old butterflies ($n$ = 37). The maximum body fat ratio of *V. cardui* is unknown, but in

monarch butterflies constitutes approximately 23% of the body mass. Applying a similar ratio to *V. cardui* leads to a maximum body fat content of 34.5 mg. The rate of fat consumption depends on the metabolic rate, which, in turn, depends on the butterfly's temperature. Monarch butterflies have resting metabolic rates of 0.4 ml·O$_2$·g$^{-1}$·h$^{-1}$ at 20 °C and 0.6 ml·O$_2$·g$^{-1}$·h$^{-1}$ at 30 °C[27]. Metabolic rates during active flight substantially exceed resting metabolic rates, ranging from 25 times higher at 32 °C[102] to 31 times higher at 22 °C[103] in monarchs. Using conversion factors of 4.7 cal·ml of O$_2$ and 0.11 mg of fat per calorie[27], we estimated fat consumption rates during active flight (0.96 mg to 1.16 mg per hour). A summary of all relevant variables is provided in Supplementary Table S5 of the Supplementary Material. We used these parameter values to estimate the fat consumption in a scenario of non-stop continuous flight across the Atlantic Ocean (Supplementary Table S6). Under all tested parameter options, non-stop continuous flight in the absence of wind requires more energy than the maximum fat content (Supplementary Table S6).

Next, we considered a scenario where the easterly trade winds preceding the capture date assisted the *V. cardui* individuals across the Atlantic. Considering the average windspeed of trajectories preceding the observation date ranged from 4.71 m/s to 8.79 m/s, and assuming a precise downwind orientation, we calculated a groundspeed during active flight within the range of 10.71 m/s to 14.79 m/s. We used the same parameter values as the previous scenario to estimate the fat consumption of non-stop continuous flight across the Atlantic Ocean with wind assistance at different speeds. Although less energy was needed than in the first scenario, non-stop continuous flight with wind assistance still requires more energy than is provided by the assumed maximum fat content (Supplementary Table S6).

Therefore, it is very unlikely that the dispersing *V. cardui* maintained consistent active flight throughout their journey. Instead, they likely also engaged in "minimum-effort" flight, wherein the butterfly generates the minimum lift necessary to remain airborne, preserving their energy. Monarchs have been observed combining active and soaring flight in a proportion of 15–85%, respectively[104]. We used this proportion to estimate the average metabolic rate for the whole flight, therefore modeling an alternating strategy of minimum-effort and active flight phases: 0.15*(active metabolic rate, AMR) + 0.85*(resting metabolic rate, RMR). This estimation assumes the RMR for the minimum-effort phase, but a slightly higher metabolic rate may be more plausible because of likely minimal flapping to maintain height. In our model, we estimate that a rate five times higher than the RMR would render the transatlantic crossing energetically unfeasible. For phases of minimum-effort flight, the butterfly's contribution to horizontal speed was assumed to be negligible. Consequently, the groundspeed during this phase is assumed to be equal to the average windspeed. To estimate the time required for the transatlantic crossing, we maintained the proportions of active and minimal-effort flight (15% and 85%) and computed the duration by dividing the minimum distance of the crossing (4200 km) by the weighted mean of the active flight speed and the minimum-effort flight speed. Under this scenario, the *V. cardui* individuals could have completed the transatlantic crossing with an initial fat content as low as 13.70%. We estimate that the time required to complete the transatlantic crossing ranged between approximately 5–8 days.

### Reporting summary

Further information on research design is available in the Nature Portfolio Reporting Summary linked to this article.

## Data availability

Demultiplexed and adapter-trimmed RAD-sequencing reads have been deposited in the European Nucleotide Archive project PRJEB57763. Raw metabarcoding sequences have been deposited in the European Nucleotide Archive project PRJEB57731. Source Data for Figs. 1 and 4 can be obtained using the provided code (see Code availability). Source Data for Fig. 2 are provided as a Source Data file. Source Data for Fig. 3 is available in Supplementary Table S3. Source data are provided with this paper.

## Code availability

Scripts used to process the data: https://github.com/GTlabIBB/Guyane (https://doi.org/10.5281/zenodo.10901404).

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

## Acknowledgements

We are grateful to our colleagues who provided observational data or contributed samples used in this study, including B. Acosta, A. Andersen, D. Benyamini, C. Brevignon, S. Cuvelier, L. Dapporto, V. Dincă, J-Y Gallard, M. Gascoigne-Pees, E. Goudegnon, J. Hernández-Roldán, J.C. Hinojosa, R. Izquierdo, M. Janda, E. Karolinskiy, M. Khaldi, R. Khellaf, M. Kiki, M. López-Munguira, A. Marsiñach, Y. Monasterio, F. Mondaca, F. Páramo, E. Pfeiler, C. Pitteloud, A. Shapiro, N. Shapoval, M. Trujano, S. Viader and R. Vodă. We thank the Centre d'étude de la biodiversité amazonienne (CEBA, French Guiana) for supporting this research. We thank R. López-Mañas for providing useful code for the wind trajectory analyses. This work was funded by the National Geographic Society

(grant WW1-300R-18), by the British Ecological Society (grant LRB16/1015), by the PRIC grants of the Fundació Barcelona Zoo, by the grant LINKA20399 from the CSIC iLink program, by the grant 2021-SGR-01334 from the Departament de Recerca i Universitats (Generalitat de Catalunya), and by the grant PID2020-117739GA-I00 MCIN/AEI/10.13039/501100011033 to G.T.; by Grant NFRE-2018-00738 of the New Frontiers in Research Fund (Government of Canada) to C.P.B. and G.T.; by grant FPU22/02358 from the Spanish Ministry of Science, Innovation and Universities to E.T.-D; and by project PID2022-139689NB-I00 (MCIN/AEI/10.13039/501100011033 and by ERDF, EU) and by grant 2021-SGR-00420 from the Departament de Recerca i Universitats (Generalitat de Catalunya) to R.V. G.T. and N.E.P. also acknowledge the Putnam Expeditionary Fund from the Museum of Comparative Zoology (MCZ, Harvard University).

## Author contributions

Conceptualization: T.S., C.P.B., R.V., N.E.P., G.T.; Field Sampling: R.V., G.T.; Data Collection: T.S., C.P.B., M.S.R., G.T.; Data analysis: T.S., C.P.B., M.S.R., E.T-D., G.T.; Writing – original draft: T.S., C.P.B., G.T.; Writing – review and editing: T.S., C.P.B., M.S.R., E.T-D., R.V., N.E.P., G.T.; Funding: T.S., C.P.B., R.V., N.E.P., G.T.

## Competing interests

The authors declare no competing interests.
