## [Peer Review File · Nature Communications]

A trans-oceanic flight of over 4,200 km by painted lady butterfliesREVIEWER COMMENTS

Reviewer #1 (Remarks to the Author):

Referee report for Nature Communications 23 Aug 2023.

A trans-oceanic flight of 1 over 4,200 km by painted lady butterflies

[Initial version, Aug 2023.]

Suchan, T., Bataille, C.P., Reich, M.S., Toro-Delgado, E. Vila, R., Pierce, N.E., & Talavera, G.

Key results

The paper uses a variety of methods to establish that a small number of painted lady butterflies caught on the NE coast of the South American continent had arrived there from West Africa using wind transport.

Validity

The key finding is convincing because it is supported by a number of independent lines of evidence – an example of ‘triangulation’. I see no reason to doubt its general validity.

Significance

The paper is the first to document butterfly movement across the Atlantic to the S. American continent. Movements of locusts to the Caribbean islands and probably also S America have been reported (and analysed) previously – the paper should recognise this more explicitly. The paper is also significant for the variety of recently developed techniques it employs, demonstrating their utility for insect migration research and providing sufficient information for others to employ them.

Data and methodology

The Methods section describes the methods used in commendable detail. As far as I can tell, they have used state-of-the-art techniques.

Analytical approach

The details of most of the procedures and analyses are outside my expertise and I cannot assess them. The integration of the various individual results into a coherent account of where the butterflies originated and how they reached their destination reads well.

Suggested improvements

See Specific Comments below. The material on the contribution of active flight, and associated fuel requirements, needs development. Fig. S1 as provided is not print quality.

Clarity and context

The paper is well written in good English and with an appropriate structure and little repetition. There are a few places where editing is needed, and see Specific Comments below for a few suggestions for improving clarity.

References

For the areas I am familiar with, the references appear mostly appropriate but I suggest a few of additions (not mine!) in the Specific Comments below.

My expertise

Insect migration and its biometeorology (including wind transport). Animal migration more generally. Remote-sensing technologies for observing insects in flight. I do not have expertise to assess the genomic analysis/Figs 2 and 3 and only limited knowledge of isotope analysis and ENM/Fig. 4.

Specific Comments

Abstract

“First proven transatlantic crossing.” Desert locusts have been documented reaching the Caribbean: these are transatlantic flights. They probably also reached S America near French Guiana, though Rosenberg and Burt (1999) seem less than 100% certain about this.

Main text

“massive insect flocks”. “exceptional circumstances” “large flock size”. This is essentially incorrect and misrepresents the capability of radar for observing insect migration. I suggest reviewing Hu et al. (2016) and Wotton et al. (2019) and recasting this text appropriately (perhaps citing one or both of these). Entomological radars can detect migrations at quite low intensities; weather radars perhaps need higher intensities but not ‘flocks’. Avoid ‘flock’ unless you mean a discrete patch of insects (as in Westbrook and Eyster 2017) – when the term ‘cloud’ may be better. Individual radars do have limited range, especially for following individual insects, so I fully agree that they cannot record LDD movements from start to finish. Of course, a major reason radar can't help with this study is that there are no radars across the Atlantic!

“a few individual moths”. I suggest adding ‘large’.

Results

Trans-oceanic wind trajectories

Either in Fig. 1 or in the accompanying text, state how many days the trajectories indicate for a backtrack to the W African coast. Give also the speeds attained in these trajectories.

Fig. S1 has very poor graphic quality and is unsatisfactory in its present form. What does the lower part of each figure represent? (Presumably these are the 'circular histograms' referred to in Methods, though they don't look like histograms. Are they referred to anywhere?)

Geolocation of natal origins

See comment below (under Methods) on precision of isoscapes and their applicability to the specific period of the flights.

Discussion

"Beyond the methodological advance...." This paragraph needs development. See comments below on coverage of this topic in Methods.

"recurrent winds at a global scale may function as regular dispersal highways and, thus, whether windborne insect dispersal may be predictable²⁴." This seems to be confusing dispersal events that lead to colonisation/"biogeographical consequences" (24), which don't need to be either regular or predictable, and adaptive migrations (like that of painted ladies between Africa and Europe) that do. Rework. I agree the dust-deposition evidence is relevant and suggests regular transport along the route identified in this study. (I also find it impressive that you apparently found these immigrants on what I presume was your first attempt, and at a single surveyed location; this suggests they may well occur frequently. This contrast with the 1988 desert locust flight, which appears to have been an exceptional event, may be worth mentioning.) Frequent, even regular, occurrence wouldn't make such flights adaptive.

"and rare migrants from Africa are most likely to arrive in the Caribbean". I don't think this follows from the earlier part of the sentence, so put it in a new sentence? The assertion also needs support. The title of ref. 29 refers to Guyane, which is a fair way from the Caribbean: is it valid? "occurrence of *V. cardui* in Hawaii and a few historical records of the species from Western Australia". Need references? If they are covered by 31-36, OK – but the titles of those refs suggest they are about other topics.

Check the monarch flight distances (3009, 4365 km) – the flight to Mexico would be the longer of the two.

"African desert locusts (*Schistocerca gregaria*), have also been reported in the Caribbean^{39,40}". Seems a bit odd not to cite Rosenberg & Burt (1999), which I think is the authoritative reference on this event.

Methods

"remote-sensing tools as wind trajectory analyses". This is confused. Wind-trajectory analysis is not remote sensing.

I am sceptical that 'isoscapes' can be determined with the precision implied by Fig. 4A,B and Figs S4,S5. Rainfall is very variable, perhaps especially in potential origin regions in Africa. That would surely affect H directly and Sr via plant growth. Is Sr geochemistry really known to 1 km resolution across Africa, the Amazon basin, etc? This is not my area of expertise, but I recommend some clarification be provided on how reliable and precise these isoscapes are and how applicable they are specifically to the conditions of September-October 2013.

The section "Duration and speed of trans-oceanic dispersal" needs development. The following notes provide some suggestions. (It's also not clear to me that this material is "Methods", and I note there is some repetition of the material in Discussion. Perhaps just treat it once, in the Discussion.)

"In a highly conservative scenario...". Rewrite to make clearer you are proposing the insects double their body mass before departing (by storing lipid fuel) and then consume all this fuel. Is there any evidence butterflies can do this? The 30% option is a simple alternative (different value for 1 parameter) and doesn't warrant a separate paragraph.

"Under favourable winds, it is estimated that an actively-flying painted lady butterfly can reach a speed of 12.5 m/s.". This doesn't make sense. Do you mean airspeed or ground speed? The former doesn't depend on the wind being favourable; the latter is determined mainly by the wind speed and wouldn't be limited to 12.5 m/s. I think you will find the average speed of the crossing was well above 12.5 m/s.

"mixed strategy of wind-assisted active flight with passive wind-assisted dispersal". I'm not sure these two are different, because the butterflies won't stay aloft unless they keep flapping their wings. I suggest the crossing was effected primarily by wind transport with continuous active flight

to keep the insects in the airstream (and out of the ocean). If they managed a forward airspeed of 6 m/s in the downwind direction that would certainly help – but this raises the additional question, not addressed in your draft, of whether they could maintain such an orientation is far from clear. Your discussion here could make use of trajectory speeds that I suggested above you should provide when discussing Fig. 1.

“wind-assisted gliding”. “transport” might be better than “gliding”. If they stop wing-beating they will fall. I don’t know if fall rates for butterflies are available, but for small insects they are quite rapid. Gliding only makes sense if there is ascending air and the migrant can either stay in it or glide to the next region of ascent. Soaring flight, occasionally followed by glides, has been recorded for Monarchs (Gibo & Pallett 1979; Gibo 1981). Thermal lift can occur over the ocean, but I think it would disrupt the high-speed airflows needed for a 6-8 day crossing. (And it may be associated primarily with cold airflows, which would not have been the case here.) Therefore I think maintaining wing-flapping to stay aloft and being borne along on the wind is the only plausible scenario. Any glides would need to be followed by more energetic flight to regain altitude. Merely maintaining height, without moving forward, will likely save energy; this is what small migrants like aphids do. But whether butterflies can do this, and how much energy would be saved, I don’t know. Note also that forward flight is only useful if the insect is oriented downwind. It’s worth noting that fuel insufficiency has also been identified for desert locusts crossing to the Caribbean and reaching the British Isles (Rosenberg and Burt 1999). One implication is that only a few exceptionally fuel-loaded and fit individuals will make it – the remainder perishing at sea.

Supp

Fig. S1 graphic quality is very poor.

“We used a global model predicting bioavailable $^{87}\text{Sr}/^{86}\text{Sr}$ at a 1 km² resolution”. Don’t you need to identify this model? Name and reference. What is it based on? (See also comments on Methods, above.)

References (for this review, and if not in MS)

Gibo, D.L. and Pallett, M.J. (1979) Soaring flight of monarch butterflies, *Danaus plexippus* (Lepidoptera: Danaidae), during the late summer migration in southern Ontario. *Canadian Journal of Zoology* 57, 1393-1401.

Hu, G., K. S. Lim, N. Horvitz, S. J. Clark, D. R. Reynolds, N. Sapir, and J. W. Chapman. 2016.

"Mass Seasonal Bioflows of High-flying Seasonal Migrants." *Science* 354: 1584-1587. doi:10.1126/science.aah4379.

Rosenberg, J. and Burt, P.J.A. (1999) Windborne displacements of desert locusts from Africa to the Caribbean and South America. *Aerobiologia* 15, 167-175.

Westbrook, J.K.; Eyster, R.S. Doppler weather radar detects emigratory flights of noctuids during a major pest outbreak. *Remote Sens. Appl. Soc. Environ.* 2017, 8, 64–70, doi:10.1016/j.rsase.2017.07.009.

Wotton, K. R., B. Gao, M. H. M. Menz, R. K. A. Morris, S. G. Ball, K. S. Lim, D. R. Reynolds, G. Hu, and J. W. Chapman. 2019. "Mass Seasonal Migrations of Hoverflies Provide Extensive Pollination and Crop Protection Services." *Current Biology* 29: 2167-2173. doi:10.1016/j.cub.2019.05.036.

Reviewer identity: V. Alistair Drake, Univ. of NSW and Univ. of Canberra, Australia.

Reviewer #2 (Remarks to the Author):

Long-range insect migration plays very important role for worldwide biodiversity, but the extent of aerial flows of insects circulating remains enigmatic because of methodological challenges. This manuscript entitled “a trans-oceanic flight of over 4,200 km by painted lady butterflies” gives an important conclusion on the first proven transatlantic crossing by *Vanessa cardui* butterflies spanning at least 4,200 km from Africa to South America and lasting at least 6-8 days.

It looks to me, this conclusion mainly came from the study to three insect samples collected on October 28th, 2013, in French Guiana, by analysis using different methods such as the wind trajectory modelling, genomics, pollen metabarcoding, ecological niche modelling, and multi-isotope geolocation of natal origins. Anyway, these analyses only provide a hypothesis of long-distance migration of insects without sufficient scientific evidence, and we cannot completely rule out the possibility of bringing several insects to a distant place due to factors such as international trade. I suggest that researchers set up monitoring points on the speculated migration routes and conduct research on the collected migratory insects to obtain more evidence to directly confirm the existence of such cross-ocean migration.

Reviewer #3 (Remarks to the Author):

The authors provide evidence based on multiple techniques to show that painted lady butterflies collected on the coast of south America had their origin in Europe/Africa. They discuss their findings in the context of long-distance movements of insects and the potential to shape biogeography and future distributions. The work is noteworthy for its drawing together of multiple strands of evidence to address the extraordinary movement capabilities of insects at a global scale and I recommend it for publication.

The document is well written, I have a few suggestions below that I hope will help:

The abstract deals with what techniques you used but not what you found. In addition, some context to the suggestion that we may be underestimating transoceanic dispersal in insects would be useful. Perhaps this is this a space limitation issue?

Any way of ageing butterflies? Might older ones have come from Western Europe or is this too simplistic?

An organisational change to consider: information about the samples should go into the results section rather than the end of the introduction and a clearer description of the insects given: sex, wing damage etc. In addition, you have gathered records of painted ladies from America: a clearer treatment of this data would be useful even if only in the supporting documents... for example are the occasional occurrences of *V. cardui* mentioned consistent with southward migration dates through Europe and Africa?

Estimates for self-powered flight distances should appear in the results rather than appearing first in the discussion.

What are the range of wind speeds in the back trajectories? As I understand these ran for 200 hours and do not include any contribution made by self-powered flight. It may be useful to expand on this further (perhaps combining the wind dispersal and self-powered flight results sections)... is self-powered flight not important given the high wind speeds? Is there anything unusual about the wind patterns on these dates? What is the absolute minimum time given a coastal departure, winds AND self-powered flight? Do painted ladies use soaring flight?

Is there anything about southward migrants physiologically that points to them being more likely to make the crossing using the SAL than during other times?

You might consider the well characterised biogeographic distributions of migrants such as the monarch butterfly in your discussion or even that of smaller migrants with worldwide distributions.

Reviewer #4 (Remarks to the Author):

The authors discovered several individuals of *Vanessa cardui*, a migratory butterfly species known from all continents except South America and Antarctica, on a beach of French Guyana. Using several multidisciplinary approaches, the authors conducted almost a detective search of the origin of those individuals. Using stable and metal isotopes, the potential origin could be either in SE part of North America, SW Europe or Africa. Using pollen analysis, the origin could be either local (but the species does not live there) or from Africa. And using next generation sequencing, the individuals belong to Afro-European populations of the species, ruling out the dispersal from North America. The authors document the possibility of trans-Atlantic dispersal of the butterflies via wind currents.

These findings are very interesting, the documentation of direct dispersal of insect across Atlantic is very important for understanding of potentials of the insect survival as well as for understanding the distribution, i.e. that the current distribution of the species is not limited by dispersal abilities

but solely by the species ecological requirements. I did not see any flaws in the methodology, but maybe an under-representation of African plant species in available DNA databases could result in a low number of African plant species in the pollen analyses. The study is original and novel in its integration of various tools, the results are of a high interest, the multidisciplinary approach will attract attention of other researchers. The authors provided all necessary details.

RESPONSE TO REVIEWERS' COMMENTS

Reviewer #1 (Remarks to the Author):

Referee report for Nature Communications 23 Aug 2023.

A trans-oceanic flight of 1 over 4,200 km by painted lady butterflies

[Initial version, Aug 2023.]

Suchan, T., Bataille, C.P., Reich, M.S., Toro-Delgado, E. Vila, R., Pierce, N.E., & Talavera, G.

Key results

The paper uses a variety of methods to establish that a small number of painted lady butterflies caught on the NE coast of the South American continent had arrived there from West Africa using wind transport.

Validity

The key finding is convincing because it is supported by a number of independent lines of evidence – an example of ‘triangulation’. I see no reason to doubt its general validity.

Significance

The paper is the first to document butterfly movement across the Atlantic to the S. American continent. Movements of locusts to the Caribbean islands and probably also S America have been reported (and analysed) previously – the paper should recognise this more explicitly. The paper is also significant for the variety of recently developed techniques it employs, demonstrating their utility for insect migration research and providing sufficient information for others to employ them.

Data and methodology

The Methods section describes the methods used in commendable detail. As far as I can tell, they have used state-of-the-art techniques.

Analytical approach

The details of most of the procedures and analyses are outside my expertise and I cannot assess them. The integration of the various individual results into a coherent account of where the butterflies originated and how they reached their destination reads well.

Suggested improvements

See Specific Comments below. The material on the contribution of active flight, and associated fuel requirements, needs development. Fig. S1 as provided is not print quality.

Clarity and context

The paper is well written in good English and with an appropriate structure and little repetition. There are a few places where editing is needed, and see Specific Comments below for a few suggestions for improving clarity.

References

For the areas I am familiar with, the references appear mostly appropriate but I suggest a few of additions (not mine!) in the Specific Comments below.

My expertise

Insect migration and its biometeorology (including wind transport). Animal migration more generally. Remote-sensing technologies for observing insects in flight. I do not have expertise to assess the genomic analysis/Figs 2 and 3 and only limited knowledge of isotope analysis and ENM/Fig. 4.

RESPONSE: We truly appreciate the dedicated comments and suggestions made by the reviewer, V. Alistair Drake. We are convinced that these insightful comments substantially improved our study. Please see point-by-point comments below.

Specific Comments

Abstract

“First proven transatlantic crossing.” Desert locusts have been documented reaching the Caribbean: these are transatlantic flights. They probably also reached S America near French Guiana, though Rosenberg and Burt (1999) seem less than 100% certain about this.

RESPONSE: We apologize for not including Rosenberg and Burt (1999) in the list of references. This is indeed a relevant citation for our study and we cite it accordingly now. Please note, however, that we refer to locusts reports in the Caribbean and South America by citing two other studies, one previous to Rosenberg and Burt (1999):

- Lorenz, M. W. Migration and trans-Atlantic flight of locusts. *Quaternary International* **196**, 4–12 (2009).
- Richardson, C. & Nemeth, D. J. Hurricane-borne African Locusts (*Schistocerca gregaria*) on the Windward Islands. *GeoJournal* **23**, 349–357 (1991).

Rosenberg and Burt, as well the other two cited studies, suggest transatlantic dispersal by locusts, providing convincing evidence because 1) Desert Locusts do not occur in America, and 2) wind trajectory analyses perfectly match occurrences at origin, midway and destination. However, trajectory analyses do not provide direct individual-based evidence about the origin of these locusts. Therefore, we argue that those indirect evidences are not sufficient to unambiguously demonstrate a transatlantic flight.

This is an important nuance related to the novelty of our article: We contribute individual-based evidence for a transatlantic flight through pollen, isotope and genetic evidence, in addition to more typically used indirect evidences as wind trajectories and presence-absence observation patterns.

So, when we refer to “first proven transatlantic crossing” we are still convinced that this is the first case of unambiguous demonstration that an individual found in point B (America) has previously been in point A (Europe/Africa). Nevertheless, we suggest a change in the abstract to avoid a misinterpretation and the sentence now reads: “... and the first verified transatlantic crossing”.

Main text

“massive insect flocks”. “exceptional circumstances” “large flock size”. This is essentially incorrect and misrepresents the capability of radar for observing insect migration. I suggest reviewing Hu et al. (2016) and Wotton et al. (2019) and recasting this text appropriately (perhaps citing one or both of these). Entomological radars can detect migrations at quite low intensities; weather radars perhaps need higher intensities but not ‘flocks’. Avoid ‘flock’ unless you mean a discrete patch of insects (as in Westbrook and Eyster 2017) – when the term ‘cloud’ may be better. Individual radars do have limited range, especially for following individual insects, so I fully agree that they cannot record LDD movements from start to finish. Of course, a major reason radar can't help with this study is that there are no radars across the Atlantic!

RESPONSE: We thank the reviewer for highlighting these inaccuracies. We have adapted the text following these suggestions. The text now reads:

Miniaturized VHF radio transmitters have been successfully used to study the navigation ability of large insects overnight¹² and radar tracking of insects is feasible over short distances¹⁰⁻¹². While promising, these technological advances limit dispersal records to short timescales (< day), sites with preexisting infrastructure (e.g., weather radar), and, due to body mass requirements, large insects. Consequently, they are not scalable to most migratory insects and sampling locations, potentially leading to an underestimation of the dispersal capacity of insects.

“a few individual moths”. I suggest adding ‘large’.

RESPONSE: Done

Results

Trans-oceanic wind trajectories

Either in Fig. 1 or in the accompanying text, state how many days the trajectories indicate for a backtrack to the W African coast. Give also the speeds attained in these trajectories.

RESPONSE: We have added a new table including the mean +- SE of the number of days required by the trajectories to reach the West African Coast (Table S5). We also represent the intersecting time of the trajectories in Figures 1 and S1.

Additionally, we have computed mean speeds for each trajectory, and present the values in a new table (Table S6) and in Figures 1 and S1. The new plots indicate the mean +- SE windspeed along the trajectory.

Fig. S1 has very poor graphic quality and is unsatisfactory in its present form. What does the lower part of each figure represent? (Presumably these are the ‘circular histograms’ referred to in Methods, though they don't look like histograms. Are they referred to anywhere?)

RESPONSE: We have improved the quality of Fig. S1. We have also added a reference to the circular histograms when discussing the trajectory results. The appearance of the histograms was due to the large number of bins (360, one per degree). For a better representation, we have reduced the number of bins to 72 (5 degrees per bin) and added a central circular space. Plots have been updated for Figure S1, where we have also included a profile of the trajectories by altitude, including its standard error.

Geolocation of natal origins

See comment below (under Methods) on precision of isoscapes and their applicability to the specific period of the flights.

RESPONSE: We provide below an extensive explanation on the accuracy of the isoscapes used in this study. To fully satisfy the reviewer, we re-run geographic assignments using a time-specific isoscape (September, October) and showed very consistent natal origin areas relative to the initial study. We then rebuilt a new d2H isoscape incorporating newly published d2H data from Ghouri et al. 2024, which incorporates data from different species and time. We re-run geographic assignments using this more conservative isoscape and showed that our geolocation conclusions remain consistent.

Discussion

“Beyond the methodological advance....” This paragraph needs development. See comments below on coverage of this topic in Methods.

RESPONSE: Thank you for the helpful comments; our detailed response is below. For clarity, we have included a new section in the results regarding the energetic and flight model.

“recurrent winds at a global scale may function as regular dispersal highways and, thus, whether windborne insect dispersal may be predictable²⁴.” This seems to be confusing dispersal events that lead to colonisation/“biogeographical consequences” (24), which don’t need to be either regular or predictable, and adaptive migrations (like that of painted ladies between African and Europe) that do. Rework.

I agree the dust-deposition evidence is relevant and suggests regular transport along the route identified in this study. (I also find it impressive that you apparently found these immigrants on what I presume was your first attempt, and at a single surveyed location; this suggests they may well occur frequently. This contrast with the 1988 desert locust flight, which appears to have been an exceptional event, may be worth mentioning.) Frequent, even regular, occurrence wouldn’t make such flights adaptive.

RESPONSE: We completely agree that these flights are not adaptive, and we have made clear along the text that this is a case of butterflies caught up in a windborne dispersal event. This contrasts with migration, which is ruled by behavior and configure bidirectional cycles (which can also use winds). In *V. cardui* and other migrants, of course, involuntary LDD events may be more likely due to their frequent migratory movements. We believe that using the term “regular dispersal highways” to recurrent instances of global circulation like the SAL is correct, and that can lead to patterns of one-way dispersal events. That makes the Atlantic coast of South America and the Caribbean a predictable hotspot for eventual arrivals of African insects, when both the SAL and the insect season in Africa are favourable. Nevertheless, we have removed the term “regular” from “regular dispersal highways”, and added “which suggest the existence of a **one-way** dispersal highway for insects” for clarity.

“and rare migrants from Africa are most likely to arrive in the Caribbean”. I don’t think this follows from the earlier part of the sentence, so put it in a new sentence? The assertion also needs support. The title of ref. 29 refers to Guyane, which is a fair way from the Caribbean: is it valid?

RESPONSE: We have rephrased this sentence and included a new reference. We now refer to the Guianas and Eastern Caribbean islands.

“occurrence of *V. cardui* in Hawaii and a few historical records of the species from Western Australia”. Need references? If they are covered by 31-36, OK – but the titles of those refs suggest they are about other topics.

RESPONSE: We have included two new references describing the presence of *V. cardui* in Hawaii and Western Australia. In Australia, *V. cardui* occurrences have been reported occasionally on sand dunes, near Perth, for particular years like in November 1978 and, more recently, in November 2022, as personally communicated by Australian colleagues. The fact that the butterflies stay at coastal dunes and do not spread reminds the case described here for French Guiana, which suggests that they could also represent immigration events. The first case (1978) is reported in the citations below, which are included in the new version of the manuscript.

Braby, MF., 2000. *Butterflies of Australia. Their identification, biology and distribution.* (2 vols). CSIRO Publishing, Collingwood.

Shields O. (1992) World distribution of the *Vanessa cardui* group. *J. Lepid. Soc.* **46**, 235–238

Check the monarch flight distances (3009, 4365 km) – the flight to Mexico would be the longer of the two.

RESPONSE: We were referring to individual data from tagged individuals, but we have rephrased this section for simplicity.

“African desert locusts (*Schistocerca gregaria*), have also been reported in the Caribbean^{39,40}”. Seems a bit odd not to cite Rosenberg & Burt (1999), which I think is the authoritative reference on this event.

RESPONSE: We thank the reviewer of the suggestions. We cited the desert locust events through other citations, but missed Rosenberg & Burt (1999), which we cite now. However, please note that we highlight here that in our report we contribute comprehensive evidence on the dispersal event and the origin. Despite the locusts’ events in the Caribbean were obvious due to the absence of the species, there were no analyses of connectivity involved other than the winds and an energetic model to support feasibility of the flight.

Methods

“remote-sensing tools as wind trajectory analyses”. This is confused. Wind-trajectory analysis is not remote sensing.

RESPONSE: The reviewer is right. We have removed the reference to “remote-sensing tools”.

I am sceptical that ‘isoscapes’ can be determined with the precision implied by Fig. 4A,B and Figs S4,S5. Rainfall is very variable, perhaps especially in potential origin regions in Africa. That would surely affect H directly and Sr via plant growth. Is Sr geochemistry really known to 1 km resolution across Africa, the Amazon basin, etc? This is not my area of expertise, but I recommend some clarification be provided on how reliable and precise these isoscapes are and how applicable they are specifically to the conditions of September-October 2013.

RESPONSE:

We would like to express our gratitude to the reviewer for allowing us to clarify those crucial points. As noted by the reviewer, hydrogen isotope ($\delta^2\text{H}$) values in precipitation exhibit temporal variations, influenced by seasons, months, days, and even specific storm events. We believe that our initial study had already fully accounted for this uncertainty but to fully satisfy the reviewer we conducted a series of sensitivity analyses to underline that our results remain consistent and conservative.

We recognize that, in an ideal world, we would have generated a time and species specific $\delta^2\text{H}$ isoscape. For our case study, an optimal isoscape would be based on precipitation isoscape built with $\delta^2\text{H}$ measurements of precipitation from the Fall 2013, and calibrated with locally-raised painted ladies collected in the Fall 2013. Such a highly-specific isoscape would have had a lower uncertainty than the one we used and would have improved the precision of geographic assignments. However, obtaining such a time and species-specific dataset is not currently possible.

While building a species- and time-specific isoscape is ideal for obtaining precise geographic assignments, using short-term (e.g., monthly) isoscapes do not necessarily improve the accuracy of geographic assignment relative to using long-term (e.g., annual) isoscapes (Vander Zanden et al., 2014). We have plotted the difference between the mean annual precipitation $\delta^2\text{H}$ and the multi-year average of $\delta^2\text{H}$ precipitation for September and October (Response Figure 1A). The differences between these two precipitation isoscapes are minimal for Europe and most of the area of interest in Africa, although some areas have more substantial differences (e.g., Mauritania).

In our study, instead of building a time-specific isoscape, we build a robust long-term $\delta^2\text{H}$ isoscape predicting mean $\delta^2\text{H}$ in butterfly wings and its associated uncertainty. In that isoscape, the spatial uncertainty map has a very large range because it encompasses uncertainties related to both the precipitation isoscape as well as the relationship between $\delta^2\text{H}$ in precipitation and $\delta^2\text{H}$ in butterfly wings (Response Figure 1B). Consequently, while the estimates of natal origin generated with this isoscape are not as precise as they could be if using a more time-specific isoscape (i.e., many possible areas of natal origin), they are conservative, integrating all the components of the uncertainty.

However, we acknowledge that the relationship between $\delta^2\text{H}$ in butterfly wings and $\delta^2\text{H}$ in mean annual precipitation that we obtained to generate our uncertainty map, is derived from data collected in temperate regions and using monarch butterflies (i.e., Hobson et al. 2019). This might not be fully appropriate to represent the full scale of uncertainty associated with a different species (i.e., painted ladies) across a much broader study area incorporating tropical and temperate zones. Fortunately, we recently analyzed $\delta^2\text{H}$ in resident butterflies of different species collected in different years and months across the Afrotropics and western Palearctic (Ghouri et al., 2024). **To address the reviewer's concern thoroughly, we build a new more conservative $\delta^2\text{H}$ isoscape** using a mean annual precipitation $\delta^2\text{H}$ isoscape calibrated with known-origin data compiled from both Hobson et al (2019) and Ghouri et al. (2024). As expected, the uncertainty of this new isoscape was higher than that of the isoscape from the initial submission, which only used data from Hobson et al. 2019 (7 to 14‰ vs. 14 to 19‰). This higher uncertainty is likely due to the broad range of $\delta^2\text{H}$ values for the multiple species of butterflies collected at different months and years in Ghouri et al. (2024). We then **re-ran the geographic assignment** using this newly generated isoscape. We show that the new estimates of natal origin retain similar spatial patterns, and highlighted areas, to the maps provided in the initial submission (Response Figure 2). We have added a figure of the hydrogen isoscape to the supplementary so that interested readers can see the uncertainty surface (Response Figure 1B; Fig S4).

Response Figure 1. (A) Difference (‰) between mean annual precipitation $\delta^2\text{H}$ and the average of long-term average of precipitation $\delta^2\text{H}$ for September and October (from waterisotopes.org). Red indicates areas where the mean annual precipitation is lower than the monthly average (max. = -72‰). Blue indicates areas where the mean annual precipitation is higher than the monthly average (max. = +20‰). **(B)** Uncertainty layer of the $\delta^2\text{H}_{\text{butterfly wing}}$ isoscape (mean annual precipitation $\delta^2\text{H}$ transformed using the conversion function provided by compiling known-origin data from Hobson et al (2019) and Ghouri et al (2023). Most high uncertainty areas correspond with the areas of greatest difference in panel A.

Response Figure 2. Joint probability surfaces from dual $\delta^2\text{H}$ and $^{87}\text{Sr}/^{86}\text{Sr}$ -based geographic assignment using a global strontium isoscape (Bataille et al., 2020) and either (A) a butterfly wing hydrogen isoscape made by the conversion of a mean annual precipitation isoscape using known-origin data from Hobson et al (2019) or (B) a butterfly wing hydrogen isoscape made by the conversion of a mean annual precipitation isoscape using known-origin data from both Hobson et al (2019) and Ghouri et al (2023). The isoscape calibrated with data from Ghouri et al (2023) is more conservative; it has a lower maximum probability and highlights additional areas as the possible natal origin.

Strontium Isoscape and Geographic Assignment Precision

We also appreciate the reviewer's attention to the geographic assignments using the strontium isoscape, which warrants additional clarification. Unlike the monthly variations seen in $\delta^2\text{H}$ values in precipitation, bioavailable $^{87}\text{Sr}/^{86}\text{Sr}$ ratios remain stable over time across the landscape. When larvae develop in a given location, they are expected to have similar $^{87}\text{Sr}/^{86}\text{Sr}$ ratios throughout the year because Sr comes primarily from geological substrates and integrates at a timescale of hundreds to thousands of years.

Nevertheless, the reviewer's concern regarding the feasibility of mapping these strontium isotope patterns at a 1 km^2 scale is well-founded. In our study, we used the global isoscape produced by Bataille et al. (2020), which is calibrated using a global bioavailable strontium isotope compilation. This global isoscape represents the best estimate of bioavailable $^{87}\text{Sr}/^{86}\text{Sr}$ to date and includes a spatially explicit map of uncertainty, intended to reflect its accuracy conservatively. Bataille et al. (2020) demonstrated that this bioavailable $^{87}\text{Sr}/^{86}\text{Sr}$ isoscape was accurate and conservative in most regions. However, in some specific data-poor areas with heterogeneous, old and rare geology with very high $^{87}\text{Sr}/^{86}\text{Sr}$, Bataille et al. (2020) demonstrated that the global bioavailable $^{87}\text{Sr}/^{86}\text{Sr}$ isoscape could be less accurate. In our study, our research area encompassed both data-rich regions in Europe and North America and data-poor regions in Africa and South America, as seen in Figure 1 of Bataille et al. (2020) (Fig S4). Therefore, the reviewer is correct in pointing out that the predictions and associated uncertainties are potentially less ideal in some of these data-poor regions (e.g., Amazon Basin, Afrotropics).

However, this is unlikely to cause significant biases in our geographic assignments. The $^{87}\text{Sr}/^{86}\text{Sr}$ of the captured butterflies are within a range of 0.710 – 0.711. These ratios are the most commonly measured bioavailable $^{87}\text{Sr}/^{86}\text{Sr}$ ratios in the compilation used to create the isoscape and are therefore accurately represented in the existing isoscape. As a result, we do not expect that additional bioavailable $^{87}\text{Sr}/^{86}\text{Sr}$ data from across the study area will change the interpretation and it is unlikely that large contiguous areas of high probability will appear when adding new bioavailable data from Africa and South America. However, we agree with the reviewer that more

extensive data collection in Africa and South America and improved modeling are required to enhance uncertainty predictions across the study area, particularly in the data-poor regions. We are actively working on this endeavor, and we anticipate that more accurate global $^{87}\text{Sr}/^{86}\text{Sr}$ isoscapes will be available in the coming years. For our current study, while acknowledging its imperfections, we rely on the best available estimate of bioavailable $^{87}\text{Sr}/^{86}\text{Sr}$ across our study area.

References:

- Ghouri S, Reich MS, Lopez-Mañas R, Talavera G, Bowen G, Vila R, Talla VNK, Collins SC, Martins DJ, Bataille CP. *A hydrogen isoscape for tracing the migration of herbivorous lepidopterans across the Afro-Paleartic range*. *Rapid Communications in Mass Spectrometry* 38(3), e9675 (2024).
- Bataille, C. P., Crowley, B. E., Wooller, M. J. & Bowen, G. J. Advances in global bioavailable strontium isoscapes, *Palaeogeography, Palaeoclimatology, Palaeoecology* 555, 109849 (2020).
- Hobson, K. A., Kardynal, K. J. & Koehler, G. Expanding the Isotopic Toolbox to Track Monarch Butterfly (*Danaus plexippus*) Origins and Migration: On the Utility of Stable Oxygen Isotope ($\delta^{18}\text{O}$) Measurements, *Frontiers in Ecology and Evolution* 7, 224 (2019).
- Vander Zanden, H. B. Wunder, M. B., Hobson, K. A., Van Wilgenburg, S. L., Wassenaar, L. I., Welker, J. M., Bowen, G. J. Contrasting assignment of migratory organisms to geographic origins using long-term versus year-specific precipitation isotope maps. *Methods in Ecology and Evolution* 5, 891–900 (2014).

The section “Duration and speed of trans-oceanic dispersal” needs development. The following notes provide some suggestions. (It’s also not clear to me that this material is “Methods”, and I note there is some repetition of the material in Discussion. Perhaps just treat it once, in the Discussion.)

RESPONSE: We thank the reviewer for the suggestions provided below. We have substantially refined this section, and have added a results section dedicated to the energetic flight model. Because of additional technical details on the calculations and assumptions of the model, we find convenient to maintain a dedicated methods section.

“In a highly conservative scenario...”. Rewrite to make clearer you are proposing the insects double their body mass before departing (by storing lipid fuel) and then consume all this fuel. Is there any evidence butterflies can do this? The 30% option is a simple alternative (different value for 1 parameter) and doesn’t warrant a separate paragraph.

RESPONSE: The energetic model section has been redone. We have now updated our calculations using a maximum value of 23% of body weight as fat, (Beall, 1948, Parlin et al., 2023). This eliminates any assumptions regarding the original body mass.

- Parlin, Adam F., et al. "The cost of movement: assessing energy expenditure in a long-distant ectothermic migrant under climate change." *Journal of Experimental Biology* 226.21 (2023).
- Beall, G. (1948). The fat content of a butterfly, *Danaus plexippus* Linn., as affected by migration. *Ecology*. 29, 80-94.

“Under favourable winds, it is estimated that an actively-flying painted lady butterfly can reach a speed of 12.5 m/s.”. This doesn’t make sense. Do you mean airspeed or ground speed? The former doesn’t depend on the wind being favourable; the latter is determined mainly by the wind speed and wouldn’t be limited to 12.5 m/s. I think you will find the average speed of the crossing was well above 12.5 m/s.

RESPONSE:

The reviewer is right that the used value 12.5 m/s, might not necessarily apply in this case. We used it as a proxy for the average groundspeed of *V. cardui* during migratory movements as estimated in a different study (Stefanescu et al., 2013). We have been able to adjust this thanks to the suggestion of the reviewer to calculate mean windspeed for our trajectories, with values presented in Table S6 and represented in Fig. S1 and Fig. 1.

We compute mean wind trajectory speed, ranging between 6.20m/s and 7.47m/s (Table S6). These values are consistent with literature on the prevalent winds in this zone of the Atlantic, which achieve average speeds of 3.7–

7.3m/s (Gentilli, 1987). However, these correspond to windspeed, so it is likely that the average groundspeed of the butterflies was slightly higher. *V. cardui* has been found to achieve self-powered flight speeds (airspeed) of 6m/s, which added to the windspeeds would result in groundspeed of 12.2-13.47m/s, consistent with the value found by Stefanescu et al. (2013).

A distance of ca. 5,700km between the center of Mali and the coast of French Guiana could be then covered in 200 hours (the complete duration of the trajectories) at a groundspeed near 8m/s (5700km/200h). Using an average groundspeed of 12.5m/s during 200 hours the butterflies would have covered a distance of 9,000km, which would be sufficient to reach far into Africa. Therefore, we find that average flight speeds below 12.5m/s are consistent with the trajectory distances and durations, as they are sufficient to complete the crossing. We have adjusted all values in our new calculations, and variables used for the energetic and flight model have been summarized in a new table for clarity (Table S7). Note that we do not mean that *V. cardui* cannot fly faster at particular moments, but we also estimate that these butterflies could not energetically sustain active flight across the whole trip (see model results in Table S8).

- Gentilli, J. (1987). Trade winds. In: Climatology. Encyclopedia of Earth Science. Springer, Boston, MA. https://doi.org/10.1007/0-387-30749-4_181

“mixed strategy of wind-assisted active flight with passive wind-assisted dispersal”. I’m not sure these two are different, because the butterflies won’t stay aloft unless they keep flapping their wings. I suggest the crossing was effected primarily by wind transport with continuous active flight to keep the insects in the airstream (and out of the ocean). If they managed a forward airspeed of 6 m/s in the downwind direction that would certainly help – but this raises the additional question, not addressed in your draft, of whether they could maintain such an orientation is far from clear. Your discussion here could make use of trajectory speeds that I suggested above you should provide when discussing Fig. 1.

RESPONSE: We thank the reviewer for highlighting this confusion. We agree that the butterflies would need to flap to stay aloft unless winds were ascending. We have removed this sentence for simplicity, since the focus of this paragraph is to explain the calculation of the estimates of self-powered flight distances. Now, we simply state that the journey could not have been completed without favourable winds.

“wind-assisted gliding”. “transport” might be better than “gliding”. If they stop wing-beating they will fall. I don’t know if fall rates for butterflies are available, but for small insects they are quite rapid. Gliding only makes sense if there is ascending air and the migrant can either stay in it or glide to the next region of ascent. Soaring flight, occasionally followed by glides, has been recorded for Monarchs (Gibo & Pallett 1979; Gibo 1981). Thermal lift can occur over the ocean, but I think it would disrupt the high-speed airflows needed for a 6-8 day crossing. (And it may be associated primarily with cold airflows, which would not have been the case here.) Therefore I think maintaining wing-flapping to stay aloft and being borne along on the wind is the only plausible scenario. Any glides would need to be followed by more energetic flight to regain altitude. Merely maintaining height, without moving forward, will likely save energy; this is what small migrants like aphids do. But whether butterflies can do this, and how much energy would be saved, I don’t know. Note also that forward flight is only useful if the insect is oriented downwind.

RESPONSE: We agree with the reviewer’s reasoning, and this sentence is no longer in the text (see previous comment).

It’s worth noting that fuel insufficiency has also been identified for desert locusts crossing to the Caribbean and reaching the British Isles (Rosenberg and Burt 1999). One implication is that only a few exceptionally fuel-loaded and fit individuals will make it – the remainder perishing at sea.

RESPONSE: We have incorporated this into the manuscript.

Supp
Fig. S1 graphic quality is very poor.

RESPONSE: We have improved the resolution of this figure and updated information and legend.

“We used a global model predicting bioavailable $^{87}\text{Sr}/^{86}\text{Sr}$ at a 1 km² resolution”. Don’t you need to identify this model? Name and reference. What is it based on? (See also comments on Methods, above.)

RESPONSE: We have moved the citation (i.e., Bataille et al., 2020) from the end of the sentence to an in-line citation in the middle of the sentence to make the reference more apparent. We also altered the paragraph to clarify the description of its basis.

References (for this review, and if not in MS)

- Gibo, D.L. and Pallett, M.J. (1979) Soaring flight of monarch butterflies, *Danaus plexippus* (Lepidoptera: Danaidae), during the late summer migration in southern Ontario. *Canadian Journal of Zoology* 57, 1393-1401.
- Hu, G., K. S. Lim, N. Horvitz, S. J. Clark, D. R. Reynolds, N. Sapir, and J. W. Chapman. 2016. "Mass Seasonal Bioflows of High-flying Seasonal Migrants." *Science* 354: 1584-1587. doi:10.1126/science.aah4379.
- Rosenberg, J. and Burt, P.J.A. (1999) Windborne displacements of desert locusts from Africa to the Caribbean and South America. *Aerobiologia* 15, 167-175.
- Westbrook, J.K.; Eyster, R.S. Doppler weather radar detects emigratory flights of noctuids during a major pest outbreak. *Remote Sens. Appl. Soc. Environ.* 2017, 8, 64–70, doi:10.1016/j.rsase.2017.07.009.
- Wotton, K. R., B. Gao, M. H. M. Menz, R. K. A. Morris, S. G. Ball, K. S. Lim, D. R. Reynolds, G. Hu, and J. W. Chapman. 2019. "Mass Seasonal Migrations of Hoverflies Provide Extensive Pollination and Crop Protection Services." *Current Biology* 29: 2167-2173. doi:10.1016/j.cub.2019.05.036.

Reviewer identity: V. Alistair Drake, Univ. of NSW and Univ. of Canberra, Australia.

Reviewer #2 (Remarks to the Author):

Long-range insect migration plays very important role for worldwide biodiversity, but the extent of aerial flows of insects circulating remains enigmatic because of methodological challenges. This manuscript entitled "a trans-oceanic flight of over 4,200 km by painted lady butterflies" gives an important conclusion on the first proven transatlantic crossing by *Vanessa cardui* butterflies spanning at least 4,200 km from Africa to South America and lasting at least 6-8 days.

It looks to me, this conclusion mainly came from the study to three insect samples collected on October 28th, 2013, in French Guiana, by analysis using different methods such as the wind trajectory modelling, genomics, pollen metabarcoding, ecological niche modelling, and multi-isotope geolocation of natal origins. Anyway, these analyses only provide a hypothesis of long-distance migration of insects without sufficient scientific evidence, and we cannot completely rule out the possibility of bringing several insects to a distant place due to factors such as international trade. I suggest that researchers set up monitoring points on the speculated migration routes and conduct research on the collected migratory insects to obtain more evidence to directly confirm the existence of such cross-ocean migration.

RESPONSE: We thank the reviewer for reading our manuscript and acknowledging the importance of our efforts in exploring long-range insect movements.

We are surprised by the reviewer's contention that our uniquely designed study, fortified with an unprecedented amalgamation of evidence, fails at providing sufficient evidence to substantiate a transatlantic crossing. The reviewer suggests that our conclusion leans on speculation, proposing instead the plausibility of a human introduction. We strongly challenge the reviewer's opinion.

The possibility that our captured specimens might be attributable to human introduction appears exceedingly remote, presenting a much less plausible sequence of events compared to a natural transatlantic crossing. The collected individuals were discovered inactive and standing in a sandy shore at dawn (6 a.m.), merely a few meters from the water of an isolated beach far from any populated area. That is an intriguing time and location for a butterfly to stand. If these individuals were transported via vessels (assuming that is the reviewer's understanding), it would entail that they survived an extended period of inactivity, potentially spanning days or even weeks, within the vessel, subsequently choosing to take flight at some point, perhaps upon reaching port, to arrive at the beach before the previous sunset. Nevertheless, as said, there was no commercial harbour nearby. It seems inconceivable that multiple individuals performed such an unusual chain of events, while flock migration is a well-established phenomenon.

The typical time frames for transatlantic crossings via vessels range from 10-12 days for Cargo and Commercial Ships, 14 days for yachts, 3-4 weeks for sailings vessels, to the fastest option of 6-8 days for cruise ships. Moreover, vessels departing from West Africa, where the pollen carried by the butterflies originated, infrequently traverse towards Northeast South America. Collectively, the possibility of these individuals crossing via a vessel appears highly improbable in terms of both butterfly behavior and maritime traffic patterns. Furthermore, alternative channels for human introductions, such as planes, seem implausible given the scientific evidence that we present. It is important to emphasize that we tracked individuals, not offspring from a potentially established population, and that the species have been exceptionally rare in the region, and without records of breeding. We demonstrate that these butterflies were most likely born in Europe, which excludes the possibility that a local breeder released them.

We firmly believe that a pivotal contribution of our work lies in demonstrating how the convergence of multiple sources of evidence dismisses alternative explanations that would hold valid if only a single source of evidence were considered. For instance, solely establishing a genetic link between South America and Africa wouldn't differentiate between recent colonization/introduction or an actual individual crossing, although scientists would typically dismiss the latter due to its lower likelihood of capture. Similarly, an isotope-based geolocation analysis alone would indicate migrant individuals, but would present a range of origin probabilities spanning North America or the Old World. However, the integration of four complementary sources of evidence enables tracing movements in an unprecedented manner for insects. This novel approach represents, to our knowledge, one of the most effective means to track individual insect long-distance movements from origin to destination.

Examples of potentially limited analyses are the previously documented crossings of the Desert Locust, studied through wind trajectory modelling, along with many other investigations into long-range insect movement relying on indirect evidence to infer movements (see references below). In our study, apart from the wind trajectory evidence favoring optimal conditions for the *V. cardui* crossing, we provide evidence of these butterflies feeding on flowers in Africa and that very likely were raised in Europe. The reviewer appears to ignore all this comprehensive evidence when asserting insufficient scientific evidence.

Additionally, the reviewer recommends standardized monitoring as being the key to comprehending cross-ocean migrations. While we concur that monitoring would aid in detecting new dispersal events (as emphasized in our paper's concluding remarks, advocating for increased insect monitoring practices), it's essential to highlight that monitoring alone would not elucidate connectivity, as we do in this study. In other words, monitoring is just the first step, but according to the reviewer's critique, it would still remain an inherent challenge to discard that any found individual would actually be a consequence of a human introduction. We firmly believe that we overcome these challenges with our methodology.

- H. Jia *et al.*, Windborne migration amplifies insect-mediated pollination services. *eLife* 11, e76230
- OTUKA, A., NIIYAMA, T., & JIANG, X. (2023). Possible source and migration pathway for early-summer immigrants of the oriental armyworm, *Mythimna separata*, arriving in northern Japan. *Journal of Integrative Agriculture*.
- JIA, H. R., GUO, J. L., WU, Q. L., HU, C. X., LI, X. K., ZHOU, X. Y., & WU, K. M. (2021). Migration of invasive *Spodoptera frugiperda* (Lepidoptera: Noctuidae) across the Bohai Sea in northern China. *Journal of Integrative Agriculture*, 20(3), 685-693.
- Tu, X., Hu, G., Fu, X., Zhang, Y., Ma, J., Wang, Y., ... & Chapman, J. W. (2020). Mass windborne migrations extend the range of the migratory locust in East China. *Agricultural and Forest Entomology*, 22(1), 41-49.
- Li, X. J., Wu, M. F., Ma, J., Gao, B. Y., Wu, Q. L., Chen, A. D., ... & Hu, G. (2020). Prediction of migratory routes of the invasive fall armyworm in eastern China using a trajectory analytical approach. *Pest Management Science*, 76(2), 454-463.
- Tu, X., Hu, G., Fu, X., Zhang, Y., Ma, J., Wang, Y., ... & Chapman, J. W. (2020). Mass windborne migrations extend the range of the migratory locust in East China. *Agricultural and Forest Entomology*, 22(1), 41-49.
- JIA, H. R., GUO, J. L., WU, Q. L., HU, C. X., LI, X. K., ZHOU, X. Y., & WU, K. M. (2021). Migration of invasive *Spodoptera frugiperda* (Lepidoptera: Noctuidae) across the Bohai Sea in northern China. *Journal of Integrative Agriculture*, 20(3), 685-693.
- Riley, J. R., Reynolds, D. R., Smith, A. D., Rosenberg, L. J., Xia-nian, C., Xiao-xi, Z., ... & Hai-kou, W. (1994). Observations on the autumn migration of *Nilaparvata lugens* (Homoptera: Delphacidae) and other pests in east central China. *Bulletin of Entomological Research*, 84(3), 389-402.
- Ge, S. S., Zhang, H. W., Liu, D. Z., Lv, C. Y., Cang, X. Z., Sun, X. X., ... & Wu, K. M. (2022). Seasonal migratory activity of *Spodoptera frugiperda* (JE Smith)(Lepidoptera: Noctuidae) across China and Myanmar. *Pest Management Science*, 78(11), 4975-4982.
- Ma, J., Wang, Y. P., Wu, M. F., Gao, B. Y., Liu, J., Lee, G. S., ... & Hu, G. (2019). High risk of the fall armyworm invading Japan and the Korean Peninsula via overseas migration. *Journal of Applied Entomology*, 143(9), 911-920.
- Otuka, A., Matsumura, M., Sanada-Morimura, S., Takeuchi, H., Watanabe, T., Ohtsu, R., & Inoue, H. (2010). The 2008 overseas mass migration of the small brown planthopper, *Laodelphax striatellus*, and subsequent outbreak of rice stripe disease in western Japan. *Applied Entomology and Zoology*, 45(2), 259-266.
- Otuka, A. (2018). Migration analyses and predictions for migratory insect pests toward Japan. In *Proceedings of the 2018 International Symposium on Proactive Technologies for Enhancement of Integrated Pest Management of Key Crops (E-book)*.

Reviewer #3 (Remarks to the Author):

The authors provide evidence based on multiple techniques to show that painted lady butterflies collected on the coast of south America had their origin in Europe/Africa. They discuss their findings in the context of long-distance movements of insects and the potential to shape biogeography and future distributions. The work is noteworthy for its drawing together of multiple strands of evidence to address the extraordinary movement capabilities of insects at a global scale and I recommend it for publication.

RESPONSE: Thank you very much for your positive comments and suggestions.

The document is well written, I have a few suggestions below that I hope will help:

The abstract deals with what techniques you used but not what you found. In addition, some context to the suggestion that we may be underestimating transoceanic dispersal in insects would be useful. Perhaps this is this a space limitation issue?

RESPONSE: We are indeed strongly limited by the established lengths for the abstracts. We have, however, tried to incorporate a few more details of our results.

Any way of ageing butterflies? Might older ones have come from Western Europe or is this too simplistic?

RESPONSE: Currently, we do not have an accurate way to age butterflies. Wing wear scores are often used as a proxy for butterfly age (i.e., worn wings are attributed to older butterflies and fresh wings to younger individuals). However, we have found this proxy to be fallible (Reich et al., 2023), likely because high-altitude, migratory flight results in less wing wear than near-surface, breeding and nectaring behaviour (Korkmaz et al., 2022).

- Korkmaz, R., Rajabi, H., Eshghi, S., Gorb, S. N. & Büscher, T. H. The frequency of wing damage in a migrating butterfly. *Insect Science* 0, 1–11 (2022).
- Reich, M. S. et al. Intercontinental panmixia despite distinct migration distances in the trans-Saharan butterfly migrant *Vanessa cardui*. Preprint at <https://doi.org/10.1101/2023.12.10.569105> (2023).

An organisational change to consider: information about the samples should go into the results section rather than the end of the introduction and a clearer description of the insects given: sex, wing damage etc. In addition, you have gathered records of painted ladies from America: a clearer treatment of this data would be useful even if only in the supporting documents... for example are the occasional occurrences of *V. cardui* mentioned consistent with southward migration dates through Europe and Africa?

RESPONSE: It is a very good suggestion to incorporate a few more information on other *V. cardui* individuals observed in South America. We have done so in lines 183-187. Indeed, we find that most of the few occurrences described in the literature or in online repositories are dated from October to December, coinciding with the period when *V. cardui* is highly present in West Africa.

Estimates for self-powered flight distances should appear in the results rather than appearing first in the discussion.

RESPONSE: We thank the reviewer for this suggestion. We have now created a section in the results dedicated to the energetic and flight models, including the estimates of feasibility of the dispersal event: flight hours vs. hours until energy reserves are depleted. We have also incorporated the distance that could be covered through self-powered flight alone. Our revised analysis underscores the importance of wind assistance in facilitating this process.

What are the range of wind speeds in the back trajectories? As I understand these ran for 200 hours and do not include any contribution made by self-powered flight. It may be useful to expand on this further (perhaps combining the wind dispersal and self-powered flight results sections)... is self-powered flight not important given the high wind speeds? Is there anything unusual about the wind patterns on these dates? What is the absolute minimum time given a costal departure, winds AND self-powered flight? Do painted ladies use soaring flight?

RESPONSE: The energetic flight model section has been substantially refined We have now computed mean trajectory speeds \pm SE, which values are presented in a new Table (Table S6). Mean speeds (\pm SE) considering segments of 1h are also represented in Figure S1 and Fig. 1. Wind speeds by segments for the whole set of trajectories (from 21st October to 31st October) range from \sim 2.5m/s to almost 14m/s (9 to 50.4km/h). The mean

trajectory speed ranges from 6.20m/s to 8.33m/s (7.47 if we do not consider trajectories posterior to the butterflies' arrival to Guyana).

As the reviewer correctly states, the trajectories involve solely wind dynamics, thus excluding any contribution from self-powered flight. We have developed a more complete model encompassing flight phases when the butterfly actively augments horizontal speed in tandem with the wind speed, while also incorporating periods of minimal effort to remain airborne but assuming no significant speeding up. We modeled the duration of the journey and the longevity of the butterfly's energy reserves. The results indicate that the dispersal event is feasible, but only under a minimum-effort strategy and the most favourable conditions of windspeed and original fat content reserves. Combining these models with the wind trajectories, we estimate the duration of the journey to range between 5 and 8 days. These models are explained in the methods section "Energetic and flight model" and in the results section "Duration and speed of trans-oceanic dispersal". See new Table S8 for the summarized results.

Is there anything about southward migrants physiologically that points to them being more likely to make the crossing using the SAL than during other times?

RESPONSE: This is a very good point. Indeed, it is very likely that southwards movements of *V. cardui* are generally longer than northwards movements in spring. However, no studies to our knowledge have quantified differential fat reserves or triggering factors that might lead southward migrants to perform more demanding flights than those in spring.

You might consider the well characterised biogeographic distributions of migrants such as the monarch butterfly in your discussion or even that of smaller migrants with worldwide distributions.

RESPONSE: Thanks for the suggestion. Indeed, wide biogeographic distributions are often observed for migratory species. Higher dispersive capacity may lead to expansions and sometimes to intriguing distributions in the form of geographical disjunctions. We mention several cases to illustrate this phenomenon along the text, including examples such as *Hypolimnas misippus* or *Vanessa atalanta*. We believe that this message is covered throughout the text, given space constraints.

Reviewer #4 (Remarks to the Author):

The authors discovered several individuals of *Vanessa cardui*, a migratory butterfly species known from all continents except South America and Antarctica, on a beach of French Guyana. Using several multidisciplinary approaches, the authors conducted almost a detective search of the origin of those individuals. Using stable and metal isotopes, the potential origin could be either in SE part of North America, SW Europe or Africa. Using pollen analysis, the origin could be either local (but the species does not live there) or from Africa. And using next generation sequencing, the individuals belong to Afro-European populations of the species, ruling out the dispersal from North America. The authors document the possibility of trans-Atlantic dispersal of the butterflies via wind currents.

These findings are very interesting, the documentation of direct dispersal of insect across Atlantic is very important for understanding of potentials of the insect survival as well as for understanding the distribution, i.e. that the current distribution of the species is not limited by dispersal abilities but solely by the species ecological requirements. I did not see any flaws in the methodology, but maybe an under-representation of African plant species in available DNA databases could result in a low number of African plant species in the pollen analyses. The study is original and novel in its integration of various tools, the results are of a high interest, the multidisciplinary approach will attract attention of other researchers. The authors provided all necessary details.

RESPONSE: We thank the reviewer for the positive take. We agree that the representation of DNA barcodes for African plants remain limited, and that other taxa could pop up in the future as the databases improve.

REVIEWERS' COMMENTS

Reviewer #1 (Remarks to the Author):

See attached file.

A trans-oceanic flight of 1 over 4,200 km by painted lady butterflies

[Revised version, Feb 2024.]

Suchan, T., Bataille, C.P., Reich, M.S., Toro-Delgado, E. Vila, R., Pierce, N.E., & Talavera, G.

The authors have responded to the reviewers' comments in commendable detail. The MS seems much improved.

Specific Comments*Abstract*

I still think “first verified transatlantic crossing”, without any further qualification, is unreasonably dismissive of the desert-locust evidence (e.g. Rosenberg and Burt 1999). “first transatlantic crossing verified by multiple lines of evidence” would be OK.

Main text

OK.

*Results*Geolocation of natal origins

Transfer “Finally,” to the next subsection?

Duration and speed of trans-oceanic dispersal

Refer to table S7 or Methods in the first sentence – otherwise it just reads as unsupported assertions.

“a behavior that is known from monarchs and other butterflies”. This needs a reference.

Discussion

“isotope-based xx display zones” \$\$To here...“Beyond the methodological advance...” This paragraph needs development. See comments below on coverage of this topic in Methods.

“recurrent winds at a global scale may function as regular dispersal highways and, thus, whether windborne insect dispersal may be predictable²⁴.” This seems to be confusing dispersal events that lead to colonisation/”biogeographical consequences” (24), which don't need to be either regular or predictable, and adaptive migrations (like that of painted ladies between African and Europe) that do. Rework. I agree the dust-deposition evidence is relevant and suggests regular transport along the route identified in this study. (I also find it impressive that you apparently found these immigrants on what I presume was your first attempt, and at a single surveyed location; this suggests they may well occur frequently. This contrast with the 1988 desert locust flight, which appears to have been an exceptional event, may be worth mentioning.) Frequent, even regular, occurrence wouldn't make such flights adaptive.

“and rare migrants from Africa are most likely to arrive in the Caribbean”. I don't think this follows from the earlier part of the sentence, so put it in a new sentence? The assertion also

needs support. The title of ref. 29 refers to Guyane, which is a fair way from the Caribbean: is it valid?

“occurrence of *V. cardui* in Hawaii and a few historical records of the species from Western Australia”. Need references? If they are covered by 31-36, OK – but the titles of those refs suggest they are about other topics.

Check the monarch flight distances (3009, 4365 km) – the flight to Mexico would be the longer of the two.

“African desert locusts (*Schistocerca gregaria*), have also been reported in the Caribbean^{39,40}”. Seems a bit odd not to cite Rosenberg & Burt (1999), which I think is the authoritative reference on this event.

Methods

“remote-sensing tools as wind trajectory analyses”. This is confused. Wind-trajectory analysis is not remote sensing.

I am sceptical that ‘isoscapes’ can be determined with the precision implied by Fig. 4A,B and Figs S4,S5. Rainfall is very variable, perhaps especially in potential origin regions in Africa. That would surely affect H directly and Sr via plant growth. Is Sr geochemistry really known to 1 km resolution across Africa, the Amazon basin, etc? This is not my area of expertise, but I recommend some clarification be provided on how reliable and precise these isoscapes are and how applicable they are specifically to the conditions of September-October 2013.

The section “**Duration and speed of trans-oceanic dispersal**” needs development. The following notes provide some suggestions. (It’s also not clear to me that this material is “Methods”, and I note there is some repetition of the material in Discussion. Perhaps just treat it once, in the Discussion.)

”In a highly conservative scenario...”. Rewrite to make clearer you are proposing the insects double their body mass before departing (by storing lipid fuel) and then consume all this fuel. Is there any evidence butterflies can do this? The 30% option is a simple alternative (different value for 1 parameter) and doesn’t warrant a separate paragraph.

“Under favourable winds, it is estimated that an actively-flying painted lady butterfly can reach a speed of 12.5 m/s.”. This doesn’t make sense. Do you mean airspeed or ground speed? The former doesn’t depend on the wind being favourable; the latter is determined mainly by the wind speed and wouldn’t be limited to 12.5 m/s. I think you will find the average speed of the crossing was well above 12.5 m/s.

“mixed strategy of wind-assisted active flight with passive wind-assisted dispersal”. I’m not sure these two are different, because the butterflies won’t stay aloft unless they keep flapping their wings. I suggest the crossing was effected primarily by wind transport with continuous active flight to keep the insects in the airstream (and out of the ocean). If they managed a forward airspeed of 6 m/s *in the downwind direction* that would certainly help – but this raises the additional question, not addressed in your draft, of whether they could maintain such an orientation is far from clear. Your discussion here could make use of trajectory speeds that I suggested above you should provide when discussing Fig. 1.

“wind-assisted gliding”. “transport” might be better than “gliding”. If they stop wing-beating they will fall. I don’t know if fall rates for butterflies are available, but for small insects they

are quite rapid. Gliding only makes sense if there is ascending air and the migrant can either stay in it or glide to the next region of ascent. Soaring flight, occasionally followed by glides, has been recorded for Monarchs (Gibo & Pallett 1979; Gibo 1981). Thermal lift can occur over the ocean, but I think it would disrupt the high-speed airflows needed for a 6-8 day crossing. (And it may be associated primarily with cold airflows, which would not have been the case here.) Therefore I think maintaining wing-flapping to stay aloft and being borne along on the wind is the only plausible scenario. Any glides would need to be followed by more energetic flight to regain altitude. Merely maintaining height, without moving forward, will likely save energy; this is what small migrants like aphids do. But whether butterflies can do this, and how much energy would be saved, I don't know. Note also that forward flight is only useful if the insect is oriented downwind.

It's worth noting that fuel insufficiency has also been identified for desert locusts crossing to the Caribbean and reaching the British Isles (Rosenberg and Burt 1999). One implication is that only a few exceptionally fuel-loaded and fit individuals will make it – the remainder perishing at sea.

Supp

Fig. S1 graphic quality is very poor.

“We used a global model predicting bioavailable $^{87}\text{Sr}/^{86}\text{Sr}$ at a 1 km² resolution”. Don't you need to identify this model? Name and reference. What is it based on? (See also comments on Methods, above.)

References (for this review, and if not in MS)

- Gibo, D.L. and Pallett, M.J. (1979) Soaring flight of monarch butterflies, *Danaus plexippus* (Lepidoptera: Danaidae), during the late summer migration in southern Ontario. *Canadian Journal of Zoology* 57, 1393-1401.
- Hu, G., K. S. Lim, N. Horvitz, S. J. Clark, D. R. Reynolds, N. Sapir, and J. W. Chapman. 2016. Mass Seasonal Bioflows of High-flying Seasonal Migrants. *Science* 354: 1584-1587. doi:10.1126/science.aah4379.
- Rosenberg, J. and Burt, P.J.A. (1999) Windborne displacements of desert locusts from Africa to the Caribbean and South America. *Aerobiologia* 15, 167-175.
- Westbrook, J.K.; Eyster, R.S. Doppler weather radar detects emigratory flights of noctuids during a major pest outbreak. *Remote Sens. Appl. Soc. Environ.* **2017**, 8, 64–70, doi:10.1016/j.rsase.2017.07.009.
- Wotton, K. R., B. Gao, M. H. M. Menz, R. K. A. Morris, S. G. Ball, K. S. Lim, D. R. Reynolds, G. Hu, and J. W. Chapman. 2019. Mass Seasonal Migrations of Hoverflies Provide Extensive Pollination and Crop Protection Services. *Current Biology* 29: 2167-2173. doi:10.1016/j.cub.2019.05.036.

Reviewer identity: V. Alistair Drake, Univ. of NSW and Univ. of Canberra, Australia.

Reviewer #3 (Remarks to the Author):

Authors have addressed my comments and I recommend the article for publication.

RESPONSE TO REVIEWERS' COMMENTS

Reviewer #1 (Remarks to the Author):

See attached file.

RESPONSE: See detailed responses below

Reviewer #3 (Remarks to the Author):

Authors have addressed my comments and I recommend the article for publication.

RESPONSE: We thank the reviewer for the insightful comments during the review process.

Referee report for *Nature Communications*

28 Feb 2024.

A trans-oceanic flight of over 4,200 km by painted lady butterflies

[Revised version, Feb 2024.]

Suchan, T., Bataille, C.P., Reich, M.S., Toro-Delgado, E. Vila, R., Pierce, N.E., & Talavera, G.

The authors have responded to the reviewers' comments in commendable detail. The MS seems much improved. The points raised below are all minor. Overall this is an impressive multidisciplinary study, with each component executed at a high level and overall a compelling story.

RESPONSE: Thank you very much. We agree that the MS has improved thanks to the review process.

Specific Comments

Abstract

I still think "first verified transatlantic crossing", without any further qualification, is unreasonably dismissive of the desert-locust evidence (e.g. Rosenberg and Burt 1999). "first transatlantic crossing verified by multiple lines of evidence" would be OK.

RESPONSE: As suggested by the editorial team, we have rephrased to: "and potentially the first verified transatlantic crossing".

Main text

OK.

Results

Duration and speed of trans-oceanic dispersal

Wouldn't this subsection go better after Trans-oceanic wind trajectories?

RESPONSE: We think that this section is better framed after all sources of evidence of the trans-oceanic dispersal are previously explained.

Refer to table S7 or Methods in the first sentence – otherwise it just reads as unsupported assertions.

RESPONSE: Done

"a behavior that is known from monarchs and other butterflies". This needs a reference.

RESPONSE: We have added a reference

Discussion

“combination of methods...were...” -> “was”

RESPONSE: Done

“isotope-based geolocation display zones” -> “displays” (“indicates” better?)

RESPONSE: Thanks, we have used “indicates”

“On the other hand, wind trajectories, pollen identification and genomics alone would not be enough to suggest a potential European origin without the isotope data.” I don’t think I agree with that. Certainly, isotope data add to the evidence, but the other three together would be pretty convincing. Reword.

RESPONSE: Isotopes are the only source of evidence indicating a European natal origin, so we consider this sentence to be accurate. Wind trajectories and pollen provide evidence about the adult butterflies being present and originating the flight in Africa, and phylogeographic data do not distinguish between Europe and Africa, thus being uninformative in this respect. For clarity, we write now “to suggest a potential **natal** European origin”.

“but occurs all year long, which suggest the existence of a one-way dispersal highway for insects.” I suggest here, or perhaps somewhere else, you identify these easterly winds as the “North-east Trade Winds” of the northern hemisphere – which are well known to be persistent, and not particularly fast-moving – both characters evident in your work.

RESPONSE: Good suggestion, we refer now to the easterly trade winds.

“For example, among the 26...one in February.” English wrong, something missing.

RESPONSE: We have removed “one in February” as this was not relevant to the message of this sentence.

“...*flavescens* presumably migrates...” “apparently” better?

RESPONSE: Ok

“Larger insects, such as African desert locusts (*Schistocerca gregaria*), have also been reported in the Caribbean⁵⁰⁻⁵²”. The several S. American species of *Schistocerca* are believed to have originated from such movements (Rosenberg & Burt 1999): given the topic of this paragraph, this is something you might want to mention.

RESPONSE: We reference to this in the first sentence of the paragraph, citing Lovejoy et al (2006), who showed the influence of trans-atlantic dispersal in the diversification of the genus *Schistocerca* through phylogenetic evidence. As suggested, we have included a sentence to highlight this idea.

“...should help preventing biodiversity threats...” -> “to prevent”. (Actually, “prevent” isn’t quite right with “threat”. “eliminate” would be correct, but I suspect “reduce” or “mitigate” is more appropriate.)

RESPONSE: Done

Methods

“GFT”. Define this. UTC – 3?

RESPONSE: Thanks for the suggestion. We have clarified that.

The subsections “Phylogeographic assignments”, “Metabarcoding of carrying pollen grains”, and “Isotope-based geographic assignment” are outside my expertise and I cannot comment on them (but note a few points of English).

“Metabarcoding of carrying pollen grains” -> “carried” (“attached” better?)

RESPONSE: Done

“with 0.1% SDS” Already stated earlier in sentence?

RESPONSE: This has been corrected

“polled and purified” -> “pooled”?

RESPONSE: This has been corrected

“Energetic flight models”

“we calculated a groundspeed during active flight within the range of 10.71 m/s to 14.79 m/s.” You should really acknowledge that this implies/assumes precise downwind orientation, which is a fairly big assumption, perhaps especially over the sea.

RESPONSE: We acknowledge this assumption now.

I doubt that butterflies could remain aloft for 85% of the time using only resting metabolism. Soaring only keeps you aloft in updrafts – and updrafts are necessarily accompanied by downdrafts. Minimal flapping to maintain height, but not move forward (or do so relatively slowly, to minimise fuel use), is a more plausible scenario. At least under the 15/85% scenario there is some fuel to spare, so weak active flight remains plausible. This issue should really be acknowledged, not glossed over.

RESPONSE: We agree with the reviewer. We explain now this assumption in the text and have calculated that the rate for a “weak active flight” could not be more than five times higher than the RMR to allow the minimum distance of 4,200 km to be energetically feasible. The new text reads:

“This estimation assumes the RMR for the minimum-effort phase, but a slightly higher metabolic rate may be more plausible because of likely minimal flapping to maintain height. In our model, we estimate that a rate five times higher than the RMR would render the transatlantic crossing energetically unfeasible.”

Figures

Fig. 3. “present in the French Guiana” -> “present in French Guiana”

RESPONSE: Done

Supplementary file

I haven’t assessed Tables S1-S4 or Figures S2-S5 as these are outside my expertise.

Table S8 has rows “Metabolic rate...” with values of 0.4 and 0.6. This should really be “Temperature ...” with values 20 and 30. An added “Metabolic rate...” row is needed with values 25x (or is it 31x – you don’t seem to say) 0.4 and 0.6 for the first case and lower values calculated from the 15/85% weighted average for the second case.

RESPONSE: Done

Table S8 again: wouldn’t it be better to have a pair of columns for zero wind, rather than an add-on row at the end?

RESPONSE: We prefer the current arrangement of the table. Introducing a column for zero wind conditions would leave several blank cells (for the minimum-effort strategy) and would condense the information displayed in the other columns.

Reviewer identity: V. Alistair Drake, Univ. of NSW and Univ. of Canberra, Australia.